# Towards Bridging the Semantic Spaces of the One-to-Many Mapping in Cross-Modality Text-to-Video Generation

## Abstract

Despite recent advances in text-to-video generation, the role of text and video latent spaces in learning a semantically shared representation remains underexplored. In this cross-modality generation task, most methods rely on conditioning the video generation process by injecting the text representation into it, rather than exploring the implicit shared knowledge between the modalities. However, the feature-based alignment of both modalities is not straightforward, especially for the *one-to-many* mapping scenario in which one text can be mapped to several valid semantically aligned videos, a challenge that generally produces a representation collapse in the alignment phase. In this work, we investigate and give insights into how both modalities cope in a shared semantic space where each modality representation is previously learned in an unsupervised way. We explore this from a latent space learning perspective by proposing a plug-and-play framework that adopts autoencoder-based models that could be used with other representations. We show that the one-to-many case requires different alignment strategies than those commonly used in the literature, which struggle to align both modalities in a semantically shared space.

## 1 Introduction

Cross-modality video generation has recently received a lot of attention due to the impressive performance of recent video generators, making it more difficult to distinguish synthetic from real samples. However, regarding the representation learning aspect of this task, particularly when coupled with joint embedding learning, it remains unclear how both modalities cope in latent space and how feature alignment occurs across different approaches. Recent works (Girdhar et al., 2023; Maiorca et al., 2023; Theodoridis et al., 2020) focus on alignment directions in latent space but employ general approaches that do not explicitly address the *one-to-many* mapping scenario, where one input from a source modality can be mapped to $n$ different and valid outputs in a target modality.

In text-to-video generation, the nature of language enables multiple textual descriptions of a single video scene, while simultaneously, a single text description can correspond to multiple valid visual interpretations. In this context, cross-modality alignment is hindered by the one-to-many mapping problem, as a collapse process is unintentionally encouraged in training. In a general one-to-one case, one input text is trained to be associated with one output video, but in the one-to-many case, one input is associated with several cross-modality outputs. A generic training pipeline in this case encourages poor alignment between the modalities that may cause collapse to the most frequent association, a mean representation of it, or even a random and nearby output in latent space.

Despite being a challenging task, the analysis of the learned joint latent space in the generative context is underexplored. In representation learning, most methods rely on classification and retrieval tasks when dealing with a joint embedding approach (Fang et al., 2022; Girdhar et al., 2023; Xue et al., 2023) to validate the learned representation. Regarding text-to-video generators, most methods focus on solutions in which the text representation is integrated through a fusion process within the video generator (Ge et al., 2022; He et al., 2022; Ho et al., 2022; Wang et al., 2024). In these approaches, the latent representation is held in the background, as this alignment is learned implicitly in the process, with evaluation focusing primarily

on video quality. While metrics can detect poor alignment, alignment-based methods would benefit from complementary analysis of the learned joint latent space.

In this context, some works have been proposed to align and generate data from multiple modalities (Tang et al., 2023), where modality-specific models are trained from scratch to learn and regularize a semantically shared representation space. Moreover, these methods rely on large-scale data sets for training, such as WebVid-2M (Bain et al., 2021) and HD-VILA-100M (Xue et al., 2022), whose large volume of text-video pairs may mask the one-to-many mapping problem. Nevertheless, there are currently several pre-trained models available in the computer vision community for text and video, yet, to the best of our knowledge, few works leverage these pre-trained representations for feature-based cross-modal alignment. Additionally, little is known about this alignment from a latent space perspective, which could provide insights into how the modalities cope in a semantically shared space.

In particular, when using autoencoding approaches that regularize the target-modality latent space, analyzing this implicit representation could aid in understanding the alignment process. For the image modality, prior work has explored this relation, from concerns about bias (Gat et al., 2022) when analyzing the latent space, ideal latent distributions for generative models (Hu et al., 2023), to understanding cross-modality alignments for classification tasks (Maiorca et al., 2023), and methods for building joint distributions from autoencoder models (Piening & Chung, 2024; Xu et al., 2019) that enable the generation process.

In this work, we aim for a better understanding of the *one-to-many* case in text-to-video generation. We take the latent space analysis perspective to investigate the structure of the case along with how the modalities cope in pure alignment-based methods. We consider this complementary information with common video quality metrics used for the problem, for which we also make an adaptation considering the several videos associated with one input text. We consider a pipeline that leverages models trained in an unsupervised way on their respective modalities, such as text and video encoders, and aligns them in a shared representational space. We show that approaches that directly align these representations (Girdhar et al., 2023) struggle with the one-to-many mapping problem, for which we propose a progressive learning strategy for analysis and as a baseline. Furthermore, we investigate the impact of self-supervised learning methods originally designed for single-modality representation learning, such as BYOL (Grill et al., 2020), SimSiam (Chen & He, 2021), and VicReg (Bardes et al., 2022), and show their limitations when applied to cross-modal alignment. The main contributions of this work[1] include the following:

- We identify the *one-to-many* mapping scenario as a key challenge in cross-modality text-to-video generation and demonstrate its impact on feature alignment approaches. To investigate this, we adopt a latent space analysis perspective to characterize the problem's structure and to explore the relationship between text and video distributions.
- We propose a unidirectional progressive text-to-video model to analyze the *one-to-many* case, mapping text first to a shared semantic space, then to the target video distribution.
- We investigate different mapping functions between the data modalities and show their impact on the shared semantic space and how the individual video modality representations can affect the alignment.

## 2 Related Works

Text-to-video generation approaches can be divided into those that inject the text as conditioning information in the video generation process and those that aim to learn a generation pipeline by aligning the latent representations of both modalities.

### 2.1 Fusion-based text-to-video generation

In multimodal machine learning, Liang et al. (2024) categorizes fusion into two types: *fusion with abstract modalities* and *with raw modalities*. In text-to-video generation, most methods adopt the former, which considers encoders to represent each modality before applying a fusion method with the two streams of data.

---

[1]The models, checkpoints and data sets generated in this work are publicly available at http://to.be.shared.

The latter employs a fusion process at early representation learning stages, such as using the raw modalities as inputs, and is less explored in the literature.

Text conditioning in fusion-based techniques ranges from simple text injection in the generation process, such as concatenating text and video embeddings (Ge et al., 2022; Wang et al., 2024), to complex fusion methods based on attention modules or Multi-layer Perceptron (MLP) layers that generate fused text-video embeddings (He et al., 2022; Ho et al., 2022). Both text-only and fused text-video embeddings can be used at different stages or layers of the video generation process. This type of injection aims to ensure that the conditioning information is maintained in the generation and aligned with the desired video semantics. Ge et al. (2022) prepend the text embedding to the video tokens in their transformer-based video generator. Ho et al. (2022) applied MLP layers to the text embedding before adding it to each residual block of their diffusion process. He et al. (2022) concatenates the conditioning information with the latent input with the option to apply or not cross-attention layers before adding it as input to a Latent Diffusion Model (LDM).

## 2.2 Cross-modality generation based on feature-alignment

Unlike approaches that inject the conditioning information into the generation process, multi-modal latent alignment focuses on creating a shared latent space between different modalities. The alignment can support various generation scenarios, ranging from a unidirectional or one-to-one generation (Theodoridis et al., 2020) to any-to-any generation (Tang et al., 2023). In the former, a sample of a target modality $M_1^t$ is generated from an input modality $M_1$, but not the other way around. In the latter, one or multiple target modalities $M_1^t, \ldots, M_m^t$ can be generated from one or multiple input modalities $M_1, \ldots, M_n$.

In the unidirectional context, Theodoridis et al. (2020) proposed a alignment of the latent spaces of two modalities using Variational Auto-Encoders (VAEs) in two separate phases. First, a VAE model for each modality is trained to learn their respective latent spaces. In a second phase, an additional VAE is used to learn a mapping between the two modalities, forming a joint embedding space between them. The alignment is learned by minimizing the Fréchet distance (C. Dowson & V. Landau, 1982) between the distributions and is validated on food image analysis and 3D hand pose estimation. Similarly, in an any-to-any context, CoDi (Tang et al., 2023) employs a two-stage process. The first stage learns the representation of each modality using an LDM. The second stage learns a shared latent space between the modalities, in which one representation is projected onto another by also injecting the target modality in the process, and alignment is achieved through a contrastive approach.

Although these approaches implicitly support text-to-video generation, they did not explore this scenario, particularly the *one-to-many* case. Furthermore, they require joint training from scratch, without leveraging pre-trained representation models for text and video that could benefit the generation process.

Moreover, other methods have been proposed in video-language pre-training for cross-modality tasks such as video-text retrieval and video question answering (Xue et al., 2022; 2023). Xue et al. (2022) proposed a method that encodes high- and low-resolution frames separately before combining them through fusion prior to cross-modal processing. Expanding beyond video-text, Girdhar et al. (2023) proposed a representation-based alignment focused on images as the main binding modality across other modalities, excluding video. In a similar vein, Maiorca et al. (2023) presented a CoDi-like (Tang et al., 2023) approach in the text-image domain, where the decoders for target modalities are pre-trained on their respective source modalities.

Regarding the alignment process, a contrastive approach is generally used, such as InfoNCE (van den Oord et al., 2018), which was adopted in CLIP (Radford et al., 2021). Other works further explore this alignment process (Li et al., 2022; Yeh et al., 2022). DeCLIP (Li et al., 2022) uses a smaller data set of 88M pairs with self-supervised learning applied to both modalities, a multi-view cross-modality loss that extends the multi-crop transformation of Caron et al. (2020), and a nearest-neighbor alignment strategy. Yeh et al. (2022) propose the removal of the negative-positive-coupling effect in learning. Although these works propose different modality augmentations, they are not directly applicable to the *one-to-many* case. Augmenting or changing the text modality could generate multiple valid mappings for a single input, potentially mixing different semantics. Moreover, InfoNCE enforces a one-to-one match between the modalities, treating other valid pairs, such as multiple videos corresponding to the same text in the one-to-many case, as "incorrect pairings" in its batch formulation of positive and negative samples.

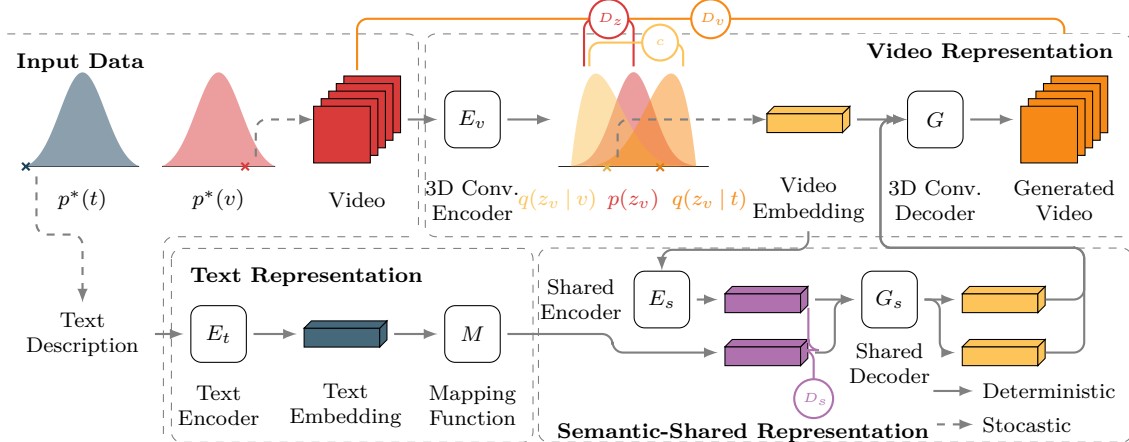

**Figure 1:** Pipeline to learn a joint semantic space $p(z)$ by bridging the gap between the conditional latent distributions of each data modality, $q(z \mid v)$ and $q(z \mid t)$, by minimizing the divergences between shared and target video representations, between the available pairs of text and video. We learn each of these posteriors individually as a variational family using the original data, $p^*(v)$ and $p^*(t)$, through a specific encoder, $E_v$ and $M \circ E_t$, respectively. Given that our evaluation task is video generation from text, we evaluate the quality of the generated videos (by decoder $G$) through a discriminator $D_v$, and by inspecting the similarity on the latent spaces of the decoupling process through additional discriminators $D_z$ and $D_s$.

## 3 Unidirectional Progressive Learning for Semantic-Shared Latent Space Alignment

To analyze and understand the one-to-many scenario, we propose a unidirectional approach for cross-modal text-to-video generation. Our pipeline consists of two phases: (1) representation learning for each modality, and (2) semantic space alignment based on progressive learning from the text modality. Through this pipeline, we analyze how alignment occurs in two key stages: the shared semantic space between modalities and the target video distribution. First, we assume that video, $v$, and text, $t$, data are in an ideal joint space, $p^*(v, t, z)$, where a latent variable, $z$, holds the joint semantic meaning of them. Hence, our problem is to learn the marginalized distribution $p^*(z)$ given the observations that come from the other marginals that are available at training time. To address the modeling problem of the semantic space $p^*(z)$, we intend to learn the marginal conditional distributions for each modality. That is, we intend to learn $p(z \mid v) \equiv p^*(z \mid v)$ and $p(z \mid t) \equiv p^*(z \mid t)$. However, learning these posteriors is intractable. Thus, we intend to approximate them with a variational family, $q(z \mid v)$ and $q(z \mid t)$, respectively, parameterized with neural networks. Finally, we need to make them similar so that the semantic information of both is equivalent.

To learn $q(z \mid v)$ and $q(z \mid t)$, we decouple each approximation into two phases. The first phase learns each modality representation, such as $q(z_v|v)$ and $q(z_t|t)$, and, the second phase learns the shared representation of $q(z \mid z_v)$ and $q(z \mid z_t)$, respectively. For $q(z_v \mid v)$, we propose a video extension of a Wasserstein Autoencoder (WAE) (Tolstikhin et al., 2018), which is trained in an unsupervised manner and, for completeness, is presented in Appendix A. Although a pre-trained model can be incorporated here as well, we select this approach to further evaluate different video architectures in the alignment process. For $q(z \mid t)$, we employ a pre-trained text encoder $E_t$, for which we do not impose any restrictions. Figure 1 presents an overview of this pipeline.

**Bridging the Semantic Spaces by Progressive Decoupling.** To learn the shared semantic space given text, $p(z \mid t)$, we approximate this posterior with a parameterized variational family, $q(z \mid t)$, for which we propose a two-part decoupling process represented by two models in hierarchical form. The first part encodes the string of words into an embedding $z_t$ using a text encoder model $E_t$. Then, $z_t$ is projected into a shared representation space with a mapping function $M : z_t \rightarrow t_s$. We map $t_s$ to the video latent space $q(z \mid t)$, through a generator $G_s$, which is the decoder part of an autoencoder model from the video latent space $q(z_v \mid v)$ to a shared representation between text and video. In this scenario, the encoder $E_s$ generates the shared representation $v_s$ from the video code sampled from $q(z_v \mid v)$.

We found empirically that trying to approximate the target distribution in a one-step approach, i.e. without a hierarchical form, led to the collapse of $q(z \mid t)$ in the *one-to-many* case. The decoupling by a hierarchical latent space is similar to Xu et al.'s (2019), but we apply to a modality that is different from the source modality. Moreover, instead of applying a regularization in the latent space in the second stage, our model applies regularization in both intermediate (shared) and target (video) latent spaces.

To link the information between the learned semantic spaces from text, $q(z \mid t)$, and video, $q(z \mid v)$, we use a WAE-based approach similar to the video semantic space, where we define:

$$D_{\mathrm{W}}\left(p^*(z), p(z \mid z_t)\right) = \inf_{q(z \mid z_v), q(z \mid z_t) \in \mathcal{Q}'} \left\{ \mathbb{E}_{z_1 \sim p^*(z)} \mathbb{E}_{z_2 \sim q(z \mid z_t)} \left[c\left(z_1, z_2\right)\right] + \lambda_{z_s} \mathcal{D}_s(q(z), p(z)) \right\}, \tag{1}$$

such that $\mathcal{Q}'$ is a non-parametric set of deterministic encoders, $z_1 \sim p^*(z)$ is a latent code representing the 'real' distribution, $q(z \mid z_v)$, $z_2 \sim q(z \mid z_t)$ is a generated latent code (through $M$) that depends on text embedding $z_t \sim q(z_t \mid t)$, and $\lambda_{z_s} > 0$ is weight for the divergence measure $\mathcal{D}_s$ between $q(z) = \mathbb{E}_{z \sim p^*(z)}\left[q(z \mid z_t)\right]$ and $p(z)$ representing our shared semantic space. For this phase, we use the cost similarity $c(z_1, z_2)$ as:

$$c(z_1, z_2) = \lambda_s \|z_1 - z_2\|_1 + \lambda_{feat}(\|G_s(z_1) - G_s(z_2)\|_1 + \|G_s(z_1) - z_v\|_1 + \|G_s(z_2) - z_v\|_1)$$
$$\lambda_{pixel}^s(\|G(G_s(z_1)) - v\|_1 + \|G(G_s(z_2)) - v\|_1), \tag{2}$$

where $G_s(z_1)$ is the video semantic code from shared code $z_1$; $G(G_s(z_1))$ is the video generated from code $G_s(z_1)$; and $\lambda_s$, $\lambda_{feat}$, and $\lambda_{pixel}^s$ are weights for shared semantic codes, video semantic space, and reconstructed videos terms, respectively.

**Shared Latent Space Divergency.** The divergence measure $\mathcal{D}_s$ is defined as:

$$\mathcal{D}_s(q(z), p(z)) = \mathcal{L}_{D_z^s} + \mathcal{L}_{bucket}, \tag{3}$$

where $\mathcal{L}_{D_z^s}$ is defined considering a shared semantic space discriminator $D_z^s$ between distribution samples $z_1$ and $z_2$ (similarly to Equation A.5), and $\mathcal{L}_{bucket}$ is a divergence loss based on a bucket approach.

In the *one-to-many* case, the text is represented by the same conditioning information, which is mapped to several semantic-related output videos, named a *bucket*. A bucket $\mathcal{B}_i$ is composed of videos with the same semantics of $t_i \in T$, such as $1 \leq i \leq N$ and $N$ is the number of different text samples. The loss $\mathcal{L}_{bucket}$ is defined with the buckets available in a training batch and follows a contrastive approach between the similarities of intra- (same semantics) and inter-bucket (different semantics) samples. Given $z_v^s \sim p(z)$ and $z_t^s \sim q(z)$, we define the loss as:

$$\mathcal{L}_{bucket}\left(z_t^s, z_v^s\right) = \frac{\lambda_{neg}}{N_t} \left( \sum_{i=1}^{N_t} \sum_{j=1, j \notin \mathcal{B}_i}^{N_v} \frac{S_{ij}}{|\overline{\mathcal{B}_i}|} \right) + \frac{\lambda_{pos}}{N_t} \left( \alpha - \sum_{i=1}^{N_t} \sum_{j=1, j \in \mathcal{B}_i}^{N_v} \lambda_{ij} \frac{S_{ij}}{|\mathcal{B}_i|} \right), \tag{4}$$

where $S_{ij} = \frac{1}{2}(\cos(z_t^s(i), z_v^s(j)) + 1)$ is the cosine similarity between embeddings $z_t^s(i)$ and $z_v^s(j)$ of the batch, $\lambda_{ij}$ is a weight for the intra-bucket pair, which is set $\lambda_{ij} = 1$ if $i \neq j$ (the sample belongs to the bucket but is not the direct match in the batch) and $\lambda_{ij} = \alpha$ if $i = j$ (direct match of the batch). The left term in Equation 4 keeps inter-bucket samples far apart, while the right term encourages intra-bucket samples to be closer. The direct match is reinforced to prevent collapse of mapping $t_i$ to the same $z_t^s$ and to maintain sample diversity.

Unlike InfoNCE loss (van den Oord et al., 2018) and its extensions (Li et al., 2022; Yeh et al., 2022), we do not treat only direct matches as positive samples in the pairwise cosine similarity phase (or diagonal match). Rather, we treat all samples within a bucket as positive samples instead of negative samples by the masking approach in Equation 4. Moreover, the bucket loss used in a one-to-one case resembles commonly used losses for contrastive learning, as each bucket will contain a single video associated with each input text. Equation 1 is the overall loss used in the progressive alignment approach, with each modality representation already trained.

# 4 Experiments

In this section, we first describe the main components of our evaluation protocol (Section 4.1), then explore the *one-to-many* case from a latent space perspective (Section 4.2). This analysis proceeds in four stages. First, we examine the general structure of the one-to-many scenario. Second, we isolate the video autoencoder model that forms the target modality representation and examine how different architectures generate different target distributions. Third, we analyze how the shared semantic space between text and video is learned, investigating how different models impact cross-modality alignment. Fourth, we present ablation experiments on the alignment process, including self-supervised methods.

## 4.1 Implementation Details

**Modality Architectures.** We use 3D convolutional deep neural networks for our probabilistic encoder $E_v$, deterministic decoder $G$, and discriminator $D_v$. For the discriminator $D_z$, mapping function $M$, video shared encoder $E_s$, and video semantic space generator $G_s$ we use fully connected networks. For video representation, we consider latent spaces with dimension $d_z$ and isotropic Gaussian prior distributions $p_z = \mathcal{N}(z; 0, \sigma^2 I_{d_z})$. We used different $d_z$ depending on the video architecture, but maintained these values in all corresponding experiments and for all data sets. We did not optimize our model for the choice of $d_z$ on any data set.

We use CLIP (Radford et al., 2021) text encoder with its pre-trained model from the ViT-B/32 version. For the video autoencoder (AE), we consider three architectures for comparison. The first is a 3D convolutional network extended from the 2D DCGAN (Radford et al., 2016) guidelines (3DConv-Base). This network does not use attention modules and residual blocks, although we added skip connections to improve its convergence. The second architecture (UNetLDM) is adapted from Rombach et al. (2022) and is based on latent diffusion. This network is extended with 3D convolutional and transposed operations and includes residual blocks and attention mechanisms. Beyond that, we also include the VDM (Ho et al., 2022) model based on diffusion to have a baseline comparison for video quality only, as this model was not proposed for representational learning with a posterior reconstruction decoder step.

**Alignment Architectures.** We use the following alignment baselines for comparison: ImageBind (Girdhar et al., 2023), CoDi (Tang et al., 2023), and CLIP (Radford et al., 2021). To focus on the alignment aspect, we extract only the alignment components from these approaches, isolating the impact of the alignment method from the modality representation learning, which varies across methods and data sets.

For the ImageBind-based alignment, we adopt its projection layer approach, which maps video and text representations to a shared semantic space. The CLIP-based alignment model employs our default mapping function architecture rather than a projection layer. Since CoDi (Tang et al., 2023) follows a different paradigm, we implement an alternative inspired by their model. Instead of using the representation injection through cross-attention layers in all the modality autoencoders, which are based on LDMs, we used the UnetLDM autoencoder with our mapping function architecture, which is based on a non-sharing representational framework, i.e., video embeddings are not passed through the text encoder and vice versa. For all baselines except CoDi, we use the 3DConv-Base video autoencoder. Additionally, all baselines employ InfoNCE (van den Oord et al., 2018) as the alignment loss.

**Data Sets and Metrics.** We consider three data sets of increasing complexity that present the one-to-many case: Moving MNIST (Mittal et al., 2017) (SyncDraw-MM), KTH Human Action (Schuldt et al., 2004), and TACoS Multi-level Corpus (Rohrbach et al., 2014). For all sets, we sampled 16-frame videos with $64 \times 64$ pixels. Additionally, we consider objective full-reference measures to evaluate video quality, which include: Peak-Signal-to-Noise Ratio (PSNR), Structural Similarity Index Measure (SSIM) (Wang et al., 2004), and perceptual metrics LPIPS (Zhang et al., 2018) and DISTS (Ding et al., 2022). Furthermore, to evaluate the generated video distributions, we consider both Fréchet Video Distance (FVD) and Kernel Video Distance (KVD) (Unterthiner et al., 2018) metrics. For text-to-video evaluation, unlike distribution-based metrics (e.g., FVD and KVD), the full-reference metrics compare against all elements in a *bucket* (the set of videos paired with the same text) rather than a single direct match, since one input text can correspond to multiple semantically aligned videos. Thus, bucket-based metrics compare a generated video to all videos in its corresponding bucket, with the final score being the best match. This approach avoids assuming only

one valid aligned video exists and instead considers the entire bucket. We also couple the evaluation with latent space analysis to better interpret the results.

**Latent Space Understanding.** For analysis, we adopt two dimensionality reduction methods: Principal Component Analysis (PCA) and t-SNE (van der Maaten & Hinton, 2008) that are focused on global and local structure preservation, respectively. For completeness, Appendices B and D also include UMAP (McInnes et al., 2018) results, another locally oriented method similar to t-SNE. Although both t-SNE and UMAP are stochastic methods, we use them to interpret the structures captured from the distributions. While different runs can generate different local structures, we focus on overall distribution relationships and complement these local views with global analysis from PCA. In our experiments, multiple runs did not result in different interpretations, as global distribution relationships remained consistent. Different runs altered individual structures but preserved the overall distribution relationships.

For the video modality, we investigate different autoencoder approaches in Section 4.2.1. As an unbiased reference outside these autoencoder-based representations for the analysis of the overall structure of the one-to-many case, we use the latent space from ViT-based embeddings obtained using VideoMAE-v2 (Wang et al., 2023), which Ge et al. (2024) demonstrated to be more effective for video representation than models commonly used for FVD computation. Other implementation details can be found in Appendix C.

### 4.2 Latent Space Understanding of the One-to-Many Scenario

In this section, we analyze the alignment of text and video modalities from a latent space perspective. We begin by examining the general structure of the latent space in this scenario, then we investigate how the target modality is learned in unsupervised manner also characterizing its structure (Section 4.2.1), and finally analyze the alignment between text and video (Section 4.2.2). To understand the impact of individual components on the overall alignment, we first isolate each component of the cross-modality alignment process, as multiple factors can contribute to successful alignment. Since we use a pre-trained model for the text modality, we focus on characterizing its structure in this initial analysis.

The visual structure of the one-to-many scenario is shown in Figure 2, with latent spaces from the CLIP text encoder and VideoMAE-v2. In the text spaces, we observe decreasing concentration from SyncDraw-MM, which has fewer, tightly concentrated clusters, to KTH and TACoS, which show more scattered distributions, with TACoS forming one large, dispersed cluster. Overall, the text modality consists of sparse, concentrated clusters in distinct regions (each representing similar semantics), while the video modality shows dense distributions for videos associated with the same text. In this regard, the data sets show decreasing one-to-many difficulty: SyncDraw-MM is most challenging, followed by KTH, with TACoS presenting the lowest difficulty.

### 4.2.1 Learning a Video Representation for the One-to-Many Scenario

The video modality can be represented using a pre-trained autoencoder, but to understand its role and the impact of different video distributions on the alignment, we trained and evaluated different architectures. Table 1 presents the quantitative results for video generation, while qualitative results are provided in Appendix C. Overall, the UNetLDM model obtained the best results across all data sets, except for TACoS, where VDM (Ho et al., 2022) achieved the best performance on FVD and KVD metrics. Notably, the UNetLDM model, which uses a backbone adapted from an LDM, generates satisfactory results without requiring the diffusion process. In contrast, while 3DConv-Base does not achieve optimal video quality, it produces satisfactory results with occasional reconstruction errors, such as confusing digits 1 and 7.

Figure 3 shows the latent spaces obtained using 3DConv-Base and UNetLDM. Although both models generate high-quality reconstructions, they exhibit different latent space distributions, with UNetLDM producing a sparser space than 3DConv-Base. Examining each space individually reveals different clustering structures, indicating that the models learn different representations for the same task despite identical training parameters and sets. The impact of these differences on alignment is explored in the next section.

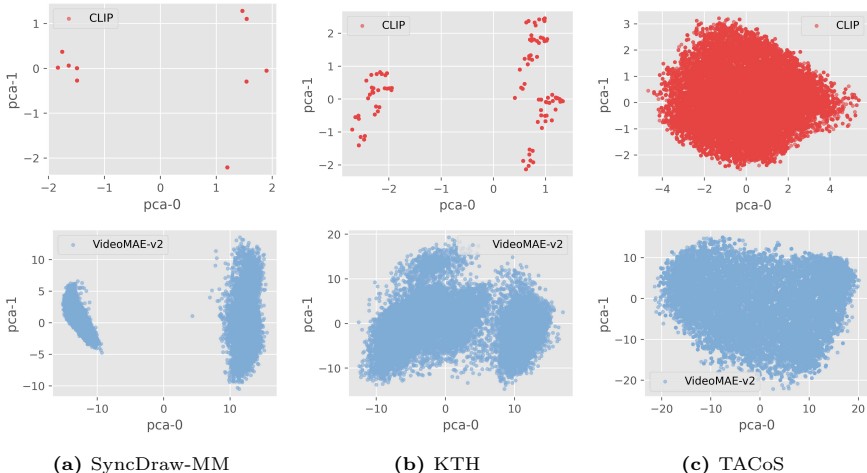

**(a)** SyncDraw-MM    **(b)** KTH    **(c)** TACoS

**Figure 2:** Visualization of the text latent spaces generated with CLIP (Radford et al., 2021) (first row) and the corresponding video latent spaces generated with VideoMAE-v2 (Wang et al., 2023) (second row), for the training split of each data set.

**Table 1:** Quantitative results of video autoencoder (AE) models on the SyncDraw-MM, KTH, and TACoS sets. Best results per column are highlighted in gray. Columns represent the corresponding metric and its values the model results over the test set. Notation: mean over the images ($\pm$ standard deviation), $\uparrow$ indicates that higher is better and $\downarrow$ that lower values are better. PSNR is in decibel scale (dB); SSIM in $[0, 1]$; LPIPS, FVD and KVD in $[0, \infty]$.

| Data Set | Model \ Metrics | PSNR↑ | SSIM↑ | LPIPS↓ | DISTS↓ | FVD↓ | KVD↓ |
|---|---|---|---|---|---|---|---|
| SyncDraw-MM | 3DConv-Base | $19.1 \pm 1.9$ | $0.89 \pm 0.03$ | $0.08 \pm 0.02$ | $0.09 \pm 0.02$ | 2.62 | 0.003 |
| | UNetLDM | $27.8 \pm 2.2$ | $0.97 \pm 0.01$ | $0.02 \pm 0.01$ | $0.03 \pm 0.01$ | 0.27 | 0.0001 |
| | VDM (Ho et al., 2022) | – | – | – | – | 4.15 | 0.006 |
| KTH | 3DConv-Base | $18.8 \pm 2.7$ | $0.41 \pm 0.16$ | $0.14 \pm 0.07$ | $0.24 \pm 0.05$ | 7.77 | 0.007 |
| | UNetLDM | $21.7 \pm 2.6$ | $0.52 \pm 0.19$ | $0.10 \pm 0.07$ | $0.20 \pm 0.06$ | 5.88 | 0.005 |
| | VDM (Ho et al., 2022) | – | – | – | – | 7.70 | 0.009 |
| TACoS | 3DConv-Base | $18.6 \pm 2.3$ | $0.54 \pm 0.07$ | $0.08 \pm 0.03$ | $0.13 \pm 0.02$ | 26.18 | 0.047 |
| | UNetLDM | $18.7 \pm 2.1$ | $0.52 \pm 0.07$ | $0.06 \pm 0.03$ | $0.13 \pm 0.02$ | 19.67 | 0.039 |
| | VDM (Ho et al., 2022) | – | – | – | – | 10.79 | 0.013 |

### 4.2.2 Learning a Semantic-Shared Representation for the One-to-Many Scenario

In this section, we analyze the alignment of the text and video modalities using our method from Section 3. For the video representation, we use the models evaluated in Section 4.2.1; for text, we use CLIP. The alignment baselines include ImageBind (Girdhar et al., 2023), CoDi (Tang et al., 2023), and CLIP (Radford et al., 2021).

Table 2 presents the quantitative results for these methods, and Figures 4 and 5 show the target video and shared semantic latent spaces obtained with them. In this section, we focus on latent space understanding, while qualitative results for these models are provided in Appendix D.2. Although the alignment approaches employ different architectures and losses, they produce similar quantitative results. No specific loss or architecture (projection layer versus progressive mapping) appears to confer a significant advantage. Moreover, the optimal model varies across data sets depending on their one-to-many complexity. However, latent space analysis reveals important differences that are not apparent in the quantitative metrics.

**Video Architectures on Alignment.** Although the UnetLDM model provides one of the best results in video reconstruction, alignment using UnetLDM does not produce the best text-to-video results. Note that both our UnetLDM-based method and the CoDi-based alternative employ UnetLDM. Quantitatively, the CoDi-based alternative yields slightly superior or equivalent results to our approach, though with different video latent space distributions. We expect a different alignment pattern since the latent space from

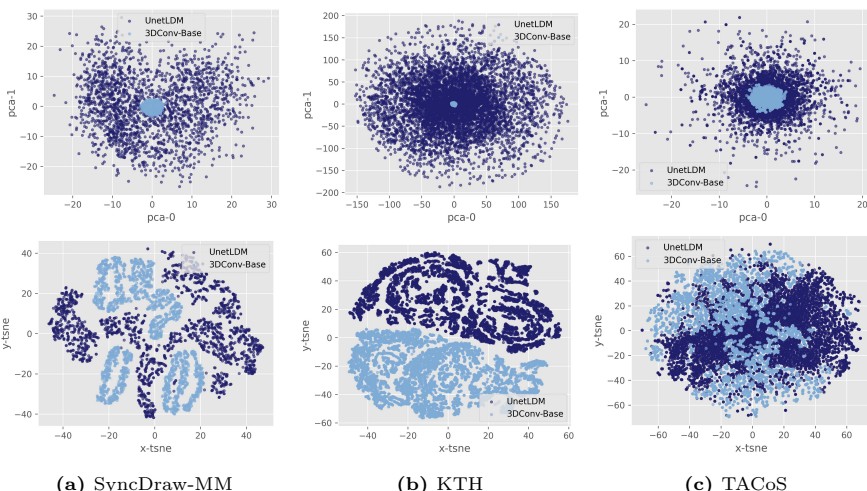

**(a)** SyncDraw-MM        **(b)** KTH        **(c)** TACoS

**Figure 3:** Visualization of video semantic spaces from 3DConv-Base and UnetLDM on the SyncDraw-MM, KTH, and TACoS data sets. For UnetLDM, we first apply PCA to reduce the dimensionality to match the latent dimension of 3DConv-Base.

**Table 2:** Quantitative results of the feature alignment between text and video modalities on three data sets: SyncDraw-MM, KTH, and TACoS. Best results per column are highlighted in gray. Notation: "B" indicates bucket approach for the metric.

| Data Set | Model \ Metrics | B-PSNR↑ | B-SSIM↑ | B-LPIPS↓ | B-DISTS↓ | FVD↓ | KVD↓ |
|---|---|---|---|---|---|---|---|
| SyncDraw-MM | CLIP-based | $20.1 \pm 1.6$ | $0.858 \pm 0.025$ | $0.25 \pm 0.06$ | $0.14 \pm 0.02$ | 7.29 | 0.007 |
| | CoDi-based | $19.9 \pm 1.3$ | $0.868 \pm 0.013$ | $0.21 \pm 0.06$ | $0.13 \pm 0.02$ | 7.19 | 0.008 |
| | ImageBind-based | $20.0 \pm 1.4$ | $0.858 \pm 0.033$ | $0.23 \pm 0.06$ | $0.13 \pm 0.02$ | 7.78 | 0.009 |
| | Our method - 3DConvBase | $19.6 \pm 1.4$ | $0.855 \pm 0.019$ | $0.21 \pm 0.05$ | $0.14 \pm 0.02$ | 6.77 | 0.006 |
| | Our method - UnetLDM | $21.2 \pm 1.6$ | $0.889 \pm 0.014$ | $0.37 \pm 0.03$ | $0.15 \pm 0.02$ | 8.18 | 0.010 |
| KTH | CLIP-based | $21.2 \pm 1.8$ | $0.367 \pm 0.094$ | $0.26 \pm 0.05$ | $0.29 \pm 0.04$ | 31.86 | 0.037 |
| | CoDi-based | $20.7 \pm 1.3$ | $0.390 \pm 0.067$ | $0.24 \pm 0.06$ | $0.29 \pm 0.05$ | 23.45 | 0.021 |
| | ImageBind-based | $19.9 \pm 2.5$ | $0.34 \ \pm 0.14$ | $0.29 \pm 0.08$ | $0.31 \pm 0.06$ | 35.78 | 0.047 |
| | Our method - 3DConvBase | $20.7 \pm 2.2$ | $0.358 \pm 0.088$ | $0.26 \pm 0.05$ | $0.31 \pm 0.04$ | 27.51 | 0.028 |
| | Our method - UnetLDM | $19.8 \pm 4.2$ | $0.28 \ \pm 0.15$ | $0.24 \pm 0.09$ | $0.30 \pm 0.08$ | 23.58 | 0.015 |
| TACoS | CLIP-based | $17.9 \pm 2.3$ | $0.508 \pm 0.081$ | $0.14 \pm 0.08$ | $0.16 \pm 0.04$ | 27.22 | 0.049 |
| | CoDi-based | $17.3 \pm 2.2$ | $0.471 \pm 0.079$ | $0.20 \pm 0.08$ | $0.19 \pm 0.04$ | 26.30 | 0.049 |
| | ImageBind-based | $17.3 \pm 2.1$ | $0.478 \pm 0.081$ | $0.18 \pm 0.07$ | $0.18 \pm 0.04$ | 29.11 | 0.051 |
| | Our method - 3DConvBase | $17.9 \pm 2.3$ | $0.505 \pm 0.083$ | $0.14 \pm 0.08$ | $0.16 \pm 0.04$ | 27.57 | 0.050 |
| | Our method - UnetLDM | $17.3 \pm 2.1$ | $0.504 \pm 0.079$ | $0.29 \pm 0.10$ | $0.22 \pm 0.05$ | 58.88 | 0.103 |

UnetLDM is sparser than the one from 3DConv-Base, as shown in Section 4.2.1, but the model seems to partially affect this result, as CoDi shows better alignment.

Furthermore, the video spaces produced by both models show a concentration of the generated video codes in particular regions that do not align with the expected distribution. For our method, even the distribution obtained with the video shared autoencoder, which maps the video latent codes to the shared semantic space, does not align with the "true" distribution (blue color). This suggests that UnetLDM has difficulties with alignment, which is partially associated with its architecture and the resulting sparser video space, as different losses are considered for this model, and the results reveal poor alignment and a level of collapse in the TACoS set (Figures 4(b) and 4(d)).

**Projection versus Progressive Approach.** Regarding projection layers versus our progressive architecture, we found that the progressive approach generates latent spaces with less collapse, although it does not fully prevent collapse on its own. Figure 4 shows more small clusters in the ImageBind-based method. Additionally, this method shows slight misalignment of the video shared codes with the expected distribution for the KTH set (Figure 4(c)). This misalignment occurs to some degree in other methods as well.

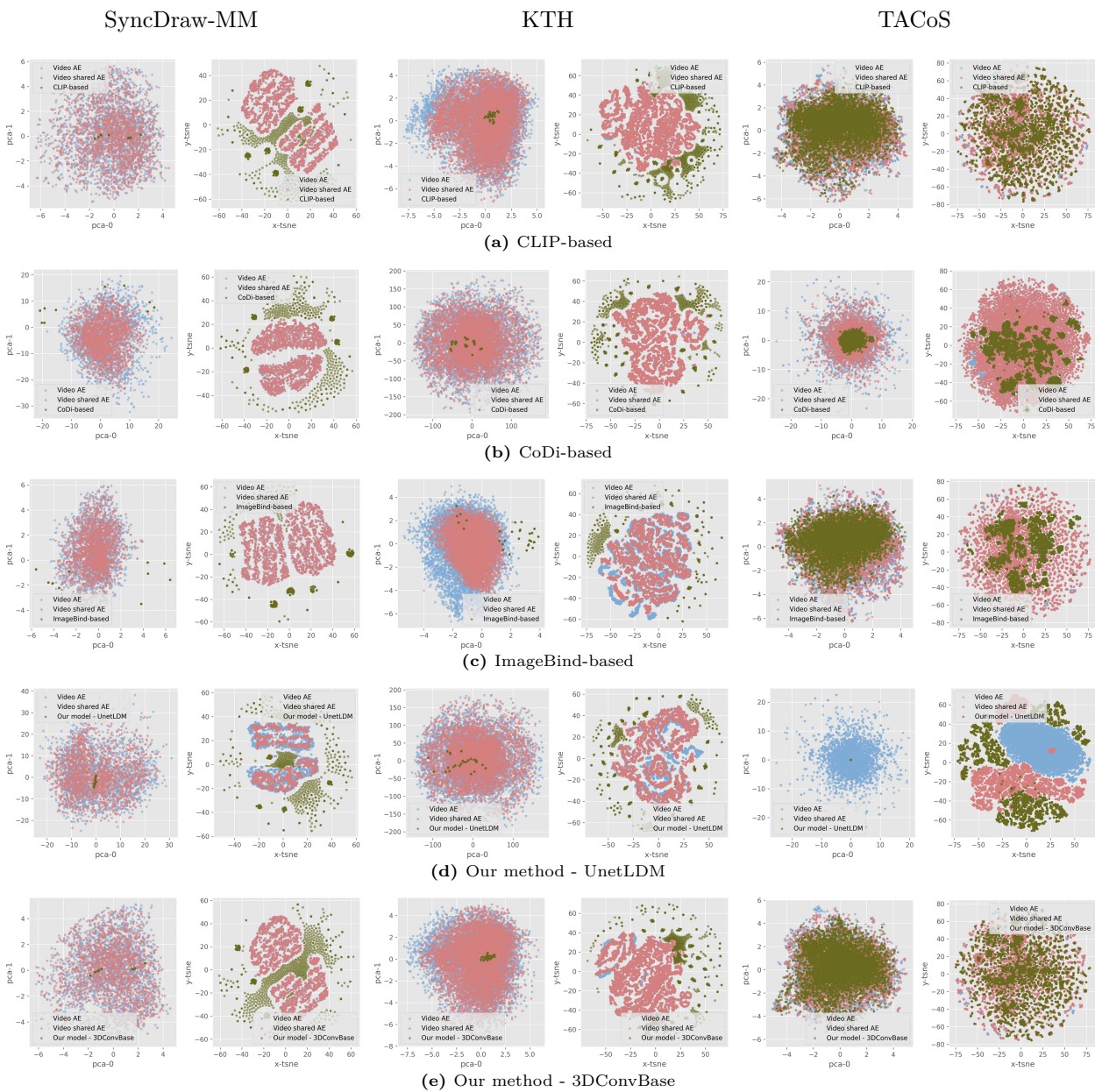

**Figure 4:** Video latent spaces from the feature alignment methods, showing embeddings from the mapping functions, video shared autoencoders, and video representation learning. Results for SyncDraw-MM (columns 1-2), KTH (3-4), and TACoS (5-6) sets using PCA (odd columns) and t-SNE (even columns).

However, when we analyze the shared semantic space in Figure 5(c), we observe similar behavior in both the projection-based model and in the TACoS set, which has lower *one-to-many* complexity. Thus, the progressive architecture appears to favor alignment compared to projection layers.

**Progressive Alignment.** While the video codes produced by the video shared autoencoder are correctly mapped to the video latent space, this does not occur entirely for the embeddings coming from the text modality, especially at higher task difficulty levels. This indicates that aligning the video latent space with the shared semantic space is more straightforward than mapping the text to the target video space. Note that in the video context, we still consider a *one-to-one* mapping, and only from the text modality this becomes our target problem.

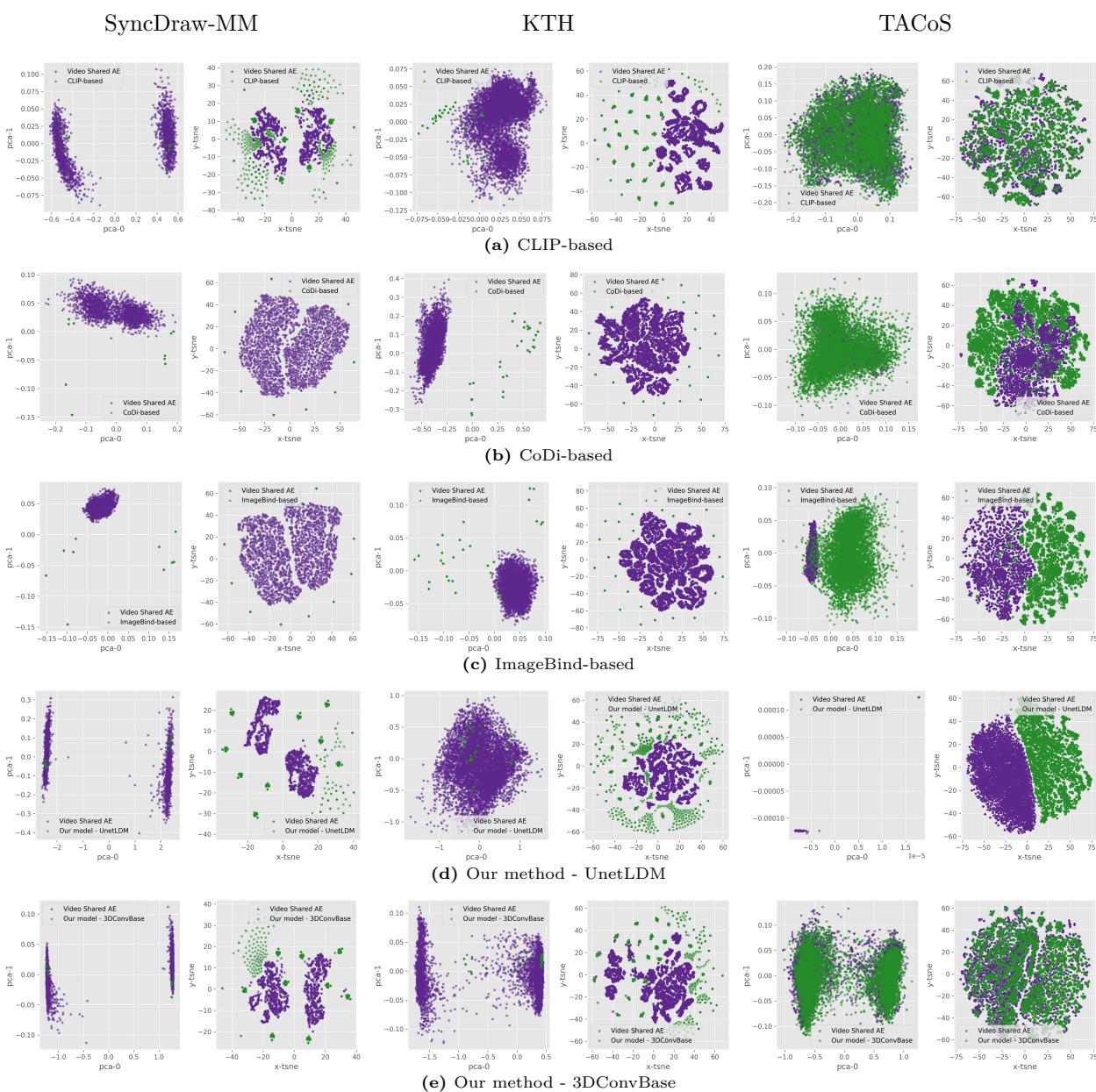

**Figure 5:** Shared semantic latent spaces from the feature alignment methods, showing embeddings from the mapping functions and video shared autoencoders. Results for SyncDraw-MM (columns 1-2), KTH (3-4), and TACoS (5-6) sets using PCA (odd columns) and t-SNE (even columns).

Analyzing the resulting shared semantic spaces in Figure 5, we observe poor alignments that propagate to the video latent space. For instance, the ImageBind-based method produces well-separated distributions, even for TACoS. CoDi also generates poor alignment for this set. The methods producing the best alignments, from a latent space perspective, appear to be CLIP-based and our method, both using 3DConv-Base. However, they still present a cluster concentration of the generated distribution, which may indicate some misalignment between the generated and expected distributions (e.g., SyncDraw-MM and KTH in Figure 4).

Moreover, the metrics used, whether bucket-based or distribution-based, are closely related to video quality. A text mapped to a video code $\hat{v}_1$ could be closer to an expected true code $v_1^t$, yet not close enough for the decoder to properly generate the expected video. A key bottleneck is the decoder's limited capacity to decode from regions near but not within the true distributions, leading to attempted decoding from points

unknown to the decoder. These cases appear to generate partially correct videos that are mostly incomplete or missing information. Thus, the metrics could not properly capture the incorrect generation, as similar incorrect decoding can generate similar metric results. Moreover, even latent space analysis, which offers an alternative perspective with valuable insights into alignment, is limited to either overall distribution analysis or manual inspection of small regions.

### 4.2.3   Ablation experiments

We conduct ablation experiments to assess the impact of our progressive approach and loss function on the alignment process. Additional ablation studies on video representation learning are included in Appendix D.1. In this section, we use the SyncDraw-MM set, where the one-to-many scenario is more prominent, and the 3DConv-Base as our video baseline. We evaluate a non-progressive method that directly aligns text with the video representation without an intermediary step, using both our adapted loss and InfoNCE (van den Oord et al., 2018). We also evaluate our progressive approach adapted to use self-supervised techniques such as VicReg (Bardes et al., 2022) and BYOL (Grill et al., 2020). We adapt these techniques by treating each augmented version as a separate modality (e.g., text and video streams rather than two views of the same modality). Since SimSiam (Chen & He, 2021) produces similar results to BYOL, we only report BYOL. Table 3 shows the quantitative results, Figures 6 and 7 shows the latent space exploration, while qualitative results are provided in Appendix D.2.

Regarding the comparison between non-progressive and progressive approaches, full-reference metrics reveal little difference in generated video quality. In contrast, distribution-based results more clearly favor the progressive method. Analysis of the learned latent spaces shows that the non-progressive mapping generates condensed clusters in regions outside the expected distributions and exhibits greater misalignment between modalities in the video space compared to the progressive mapping. Although misalignment persists, generated and expected distributions show greater proximity in progressive methods than in non-progressive methods, which produce more distant global alignments. Mapping the text to video in two phases introduces an intermediate representation (shared semantic space) that is regularized during training, unlike mapping directly with one phase. This regularization enforces alignment in the intermediate phase before generating the target distribution in the subsequent step. In contrast, one-phase alignment is more challenging because it must generate the desired distribution in a single step, with the model more constrained in performing the transformation between latent spaces.

This more distant alignment in non-progressive methods is not improved by the bucket loss from Section 3. In fact, InfoNCE performs better on distribution-based metrics, with bucket-based metrics being close to each other. The global structures of both non-progressive methods present similar relationships but different arrangements of the distributions in t-SNE visualization. However, PCA reveals generated distributions falling outside the expected regions for the non-progressive approach using InfoNCE. Since InfoNCE considers only the direct match in each text-video training pair as a positive sample, this appears to encourage misalignment outside the expected distribution. When using the bucket loss, which treats all samples from the same bucket (sharing the same input text) as positive samples, the generated distributions exhibit different behavior, though misalignment persists. The bucket loss gives greater weight to the direct match to enforce diversity within the bucket, but this diversity is not achieved with the non-progressive approach.

Considering the self-supervised approaches, VicReg and BYOL both generate similar - though not identical - structures in the video and shared semantic latent spaces, with VicReg achieving slightly better quantitative performance. We observe that their video space distributions are similar to those of our model and CLIP-based alignment (Figures 4(a) and 4(e)). However, their shared semantic spaces differ from the best-performing models in Section 4.2.2, which exhibit a split of the distribution into two regions. Additionally, VicReg and BYOL produce more compact distributions that span narrower ranges along both axes.

Furthermore, all evaluated approaches demonstrate the difficulty of aligning text and video on SyncDraw-MM and reveal that these strategies do not adequately address the one-to-many mapping challenge.

**Table 3:** Quantitative results for ablation experiments on text-video feature alignment using the SyncDraw-MM data set and the 3DConv-Base video autoencoder.

| Model/Metrics | B-PSNR↑ | B-SSIM↑ | B-LPIPS↓ | B-DISTS↓ | FVD↓ | KVD↓ |
|---|---|---|---|---|---|---|
| Non-progressive | $19.7 \pm 0.9$ | $0.856 \pm 0.014$ | $0.20 \pm 0.01$ | $0.13 \pm 0.01$ | 8.47 | 0.011 |
| Non-progressive w/ InfoNCE | $19.0 \pm 1.4$ | $0.841 \pm 0.027$ | $0.20 \pm 0.05$ | $0.14 \pm 0.02$ | 8.10 | 0.008 |
| Progressive - VicReg | $19.6 \pm 1.7$ | $0.852 \pm 0.036$ | $0.20 \pm 0.05$ | $0.13 \pm 0.02$ | 6.67 | 0.006 |
| Progressive - BYOL | $19.6 \pm 1.6$ | $0.856 \pm 0.032$ | $0.19 \pm 0.05$ | $0.13 \pm 0.02$ | 7.14 | 0.007 |

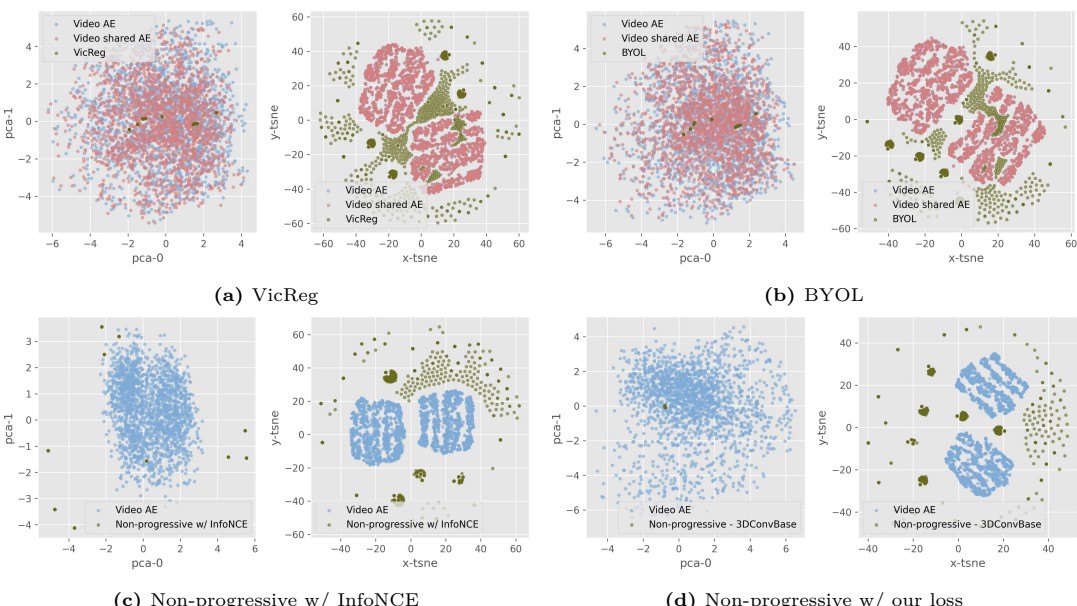

**(a)** VicReg  **(b)** BYOL

**(c)** Non-progressive w/ InfoNCE  **(d)** Non-progressive w/ our loss

**Figure 6:** Video latent spaces from ablation experiments, showing embeddings from the mapping functions, video shared autoencoders, and video representation learning, visualized with PCA (odd columns) and t-SNE (even columns).

## 5  Limitations

We explored a particular class of video architectures for representation learning based on autoencoder models. Other architectures such as the Vision Transformer (ViT) from VideoMAE (Wang et al., 2023) could be adapted with a full video decoder. Moreover, the representation learning method for the target modality could be extended to other approaches with different assumptions on the data distributions, such as relational regularization (Xu et al., 2020) and diffusion-based VampPriors (Kuzina & Tomczak, 2024), to understand their impact on the video semantic space. Furthermore, we focus on a scenario in which the video target

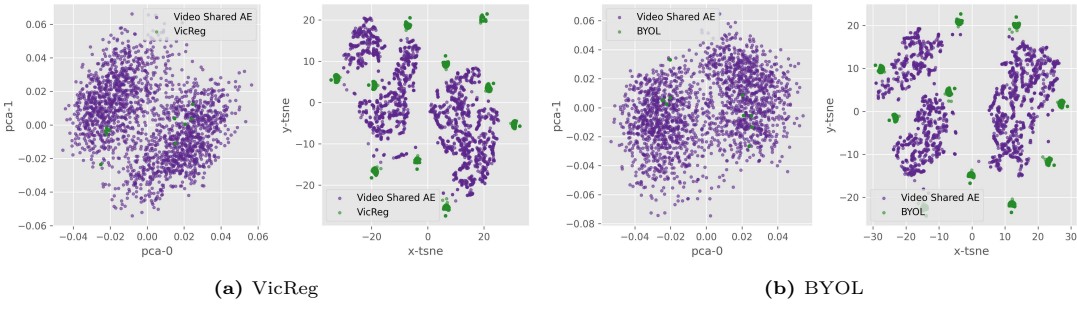

**(a)** VicReg  **(b)** BYOL

**Figure 7:** Shared semantic latent spaces from ablation experiments, showing embeddings from the mapping functions and video shared autoencoders, visualized with PCA (odd columns) and t-SNE (even columns).

modality is represented by fixed chunks of video. Future work can consider merging these chunks of video and preserving spatial and temporal consistency for a longer video generation based on a cross-modality approach with feature alignment.

## 6 Discussion and Conclusions

We shed light on an implicit problem of cross-modality generation that is currently underexplored. Although modality alignment in a shared semantic space could benefit from knowledge obtained by pre-trained models for each modality, a challenging aspect arises related to how to map both modalities. For the one-to-many case, we propose a latent space analysis perspective for assessing alignment methods based on a progressive learning framework coupled with bucket loss to learn a shared semantic space between text and video modalities. We show that the one-to-many case has different levels of complexity across different data sets and impacts the overall results of text-to-video generation from a semantic shared space. Moreover, this task lacks effective quantitative metrics for evaluation, requiring complementary methods for robust assessment.

In this work, we focus on autoencoder models as their nature implicitly enables representation learning of the modality and can be adapted for cross-modality generation by feature alignment. We show how some components of this task affect the overall result and demonstrate that video representation plays an important role in it. Overall, tackling the one-to-many case is not straightforward, requiring a different perspective when considering a semantically shared space between modalities, as current methods and regularization techniques are not designed with this case in mind.

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

## Appendix

## A    Video Semantic Space

To generate videos, we need to learn a distribution $p(v|z_v)$ that is conditioned on our semantic space and that is similar to the original video data $p^*(v)$. Toward this goal, we minimize the Wasserstein distance between both distributions by using its dual form (Bousquet et al., 2017; Tolstikhin et al., 2018) of optimizing through random encoders $q(z_v \mid v)$ instead of the original distribution couplings. Hence, we minimize

$$D_{\mathrm{W}}\left(p^*(v), p(v \mid z_v)\right) = \inf_{q(z_v \mid v) \in \mathcal{Q}} \left\{ \mathbb{E}_{x \sim p^*(v)} \mathbb{E}_{z_v \sim q(z_v \mid v)} \left[c\left(x, y\right)\right] + \lambda_z \mathcal{D}(q(z_v), p(z_v)) \right\}, \tag{A.1}$$

where $\mathcal{Q}$ is a non-parametric set of probabilistic encoders, $p(v \mid z_v)$ is our generative distribution, $x \sim p^*(v)$ is a ground truth video, $y \sim p(v \mid z_v)$ is a generated video (through decoder $G$) that depends on the semantic vector $z_v \sim q(z_v \mid v)$, and $\lambda_z > 0$ is a hyperparameter that weights the divergence measure $\mathcal{D}$ between the marginal distribution $q(z_v) = \mathbb{E}_{v \sim p^*(v)}\left[q(z_v \mid v)\right]$ and the prior $p(z_v)$ for our semantic space, and $c$ is a similarity cost.

**Video Similarity.** The cost function $c$ represents a measure between two videos, which we define as

$$c(x, y) = \lambda_{pixel}\big\|x - y\big\|_1 + \lambda_f\big\|f_{D_v}(x) - f_{D_v}(y)\big\|_1 + \lambda_p\big\|f_{\mathrm{VGG}}(x) - f_{\mathrm{VGG}}(y)\big\|_2^2, \tag{A.2}$$

where $f_{D_v}(x)$ denotes the features of an intermediate layer of the video discriminator $D_v$, when considering video $x$; similarly, $f_{\mathrm{VGG}}(x)$ denotes the features of a VGG19 network (Johnson et al., 2016); and $\lambda_{pixel} > 0$, $\lambda_f > 0$, and $\lambda_p > 0$ are hyperparameters that define the weight of each term in the final cost.

This cost function penalizes the discrepancy between the videos on the pixel (left term) and feature space (middle to right term). The penalization on the feature space acts as a perceptual similarity measure between the original and generated samples, since pixel-wise metrics have difficulties capturing perceptual properties of the reconstructed samples. Our perceptual measure is defined as a feature-matching loss (Bao et al., 2017; Salimans et al., 2016) over feature space $f_{D_v}$ of discriminator $D_v$ and feature space $f_{\mathrm{VGG}}$. We introduce the details of $D_v$ later in this section.

**Video Latent Space Divergency.** The divergence $\mathcal{D}$ represents a cost on the difference between two given spaces. In the original WAE (Tolstikhin et al., 2018), this divergence is obtained using a GAN or Maximum Mean Discrepancy approach. In contrast, we consider a metric based on feature matching (Salimans et al., 2016), which we found to be more stable to train. We convert the WAE-GAN divergence (Tolstikhin et al., 2018), defined as a non-saturating loss (Fedus et al., 2018; Goodfellow et al., 2014), into a distance minimization problem between the semantic feature spaces, $f_{D_z}$, of both $q(z_v)$ and $p(z_v)$. We empirically found that removing the min-max between the autoencoder (i.e., $E_v$ and $G$) and the discriminator $D_z$ led to a more stable training compared to the original WAE-GAN loss. Adding a gradient penalty (Fedus et al., 2018; Gulrajani et al., 2017) also leads to stable training, but we found that the feature matching term was enough to stabilize video training. Hence, we define the divergence as the aggregate

$$\mathcal{D}(q(z_v), p(z_v)) = \mathcal{L}_f + \mathcal{L}_{D_z} + \mathcal{L}_{D_v}, \tag{A.3}$$

where the losses $\mathcal{L}_{(\cdot)}$ depend on the same arguments as $\mathcal{D}$. The feature-matching loss $\mathcal{L}_f$ penalizes the semantic feature space induced by discriminator $D_z$, when it learned to distinguish between the true and a variational approximation of the semantic distributions. The video adversarial loss, $\mathcal{L}_{D_v}$, measures the similarity in the perceptual space as similar videos will have similar underlying semantic distributions, and the semantic discriminator loss, $\mathcal{L}_{D_z}$, induces similarity between prior and approximated semantic distributions.

We consider the feature-matching loss as

$$\mathcal{L}_f(q(z_v), p(z_v)) = \mathbb{E}_{\tilde{z}_v \sim p(z_v)} \mathbb{E}_{z_v \sim q(z_v)} \big\|f_{D_z}(\tilde{z}_v) - f_{D_z}(z_v)\big\|_2^2, \tag{A.4}$$

such as $f_{D_z}(z_v)$ denotes the features of an intermediate layer of $D_z$ when considering the latent vector $z_v$, and the joint semantic space $p(z_v)$ is modeled as a multivariate normal distribution.

**Table B.1:** Data set splits used for training and testing containing the number of text and video pairs along with its corresponding number of buckets.

| Model | Train | Validation | Test | Buckets |
|---|---|---|---|---|
| SyncDraw-MM | 10000 | 2000 | 2000 | 20 |
| KTH | 21030 | 5502 | 6650 | 150 |
| TACoS | 31392 | 7848 | 9811 | 11659 |

Then, we define the semantic discriminator loss to penalize the difference between the true distribution, $p(z_v)$, and our approximation, $q(z_v)$, as

$$\mathcal{L}_{D_z} = - \mathop{\mathbb{E}}_{\tilde{z}_v \sim p(z_v)} [\log D_z(\tilde{z}_v)] - \mathop{\mathbb{E}}_{z_v \sim q(z_v)} [\log(1 - D_z(z_v))], \tag{A.5}$$

where $D_z$ is the semantic space discriminator. Finally, the video discriminator $D_v$, from which we compute $f_{D_v}$ in Equation A.2, tries to differentiate between real, $p^*(v)$, and generated videos, $p(v \mid z_v)$, with a loss similar to Equation A.5 but now considering the videos samples instead of the semantic vectors.

## B  Data sets

Moving MNIST is an extension of the MNIST (Lecun et al., 1998) data set where one or two digits move up and down, left to right, and vice versa. Each video has a sentence describing the digits and their moving direction. For the KTH data set, which contains videos of several actions of 25 persons recorded in four different backgrounds with variations in light and clothing, we selected a subset of these actions (i.e., walking, jogging, and running) as in Mittal et al. (2017) and Marwah et al. (2017) experiments. We also provide a new set of text descriptions for this data[2]. Each text description indicates the person in the video, its corresponding action, and direction of movement, such as "person 2 is walking left to right" and "person 5 is jogging right to left." Lastly, the TACoS data set contains videos of people cooking with multilevel descriptions, such as one sentence, short, and detailed descriptions for each video. In our experiments, we selected the set of short descriptions that better represents the *one-to-many* case, with each description depicting an event over a time interval in the video. We present data sets splits in Table B.1 and bucket examples in Figure B.1.

## C  Implementation Details

In this section, we present additional implementation details used for latent space analysis, metrics, and model architectures.

### C.1  Latent Space Analysis

The t-SNE visualization is produced by first reducing the input dimensionality to 32 components with PCA, and then applying t-SNE over the resulting components with a perplexity of 40 and a number of iterations equal to 600 for all visualizations. Both t-SNE (van der Maaten & Hinton, 2008) and UMAP (McInnes et al., 2018) are stochastic methods with the goal of preserving local structure. In order to reproduce their initial randomness process, a random state variable can be used in these methods (e.g. seed of 42).

For the VideoMAE (Tong et al., 2022; Wang et al., 2023) representation used for the analysis of the overall structure of the one-to-many case, we selected the VideoMAE-v2 (Wang et al., 2023) model, more specifically the Hybrid-PT-SSv2-FT version used in the work of Ge et al. (2024)[3]. The ViT-g encoder features were extracted following their guidelines, generating embeddings with dimension 1408 from the penultimate layer of the encoder that were averaged across all patches.

---

[2]Mittal et al. (2017) and Marwah et al. (2017) also generated a set of text descriptions for the KTH data set, but they are not publicly available.
[3]https://github.com/songweige/content-debiased-fvd

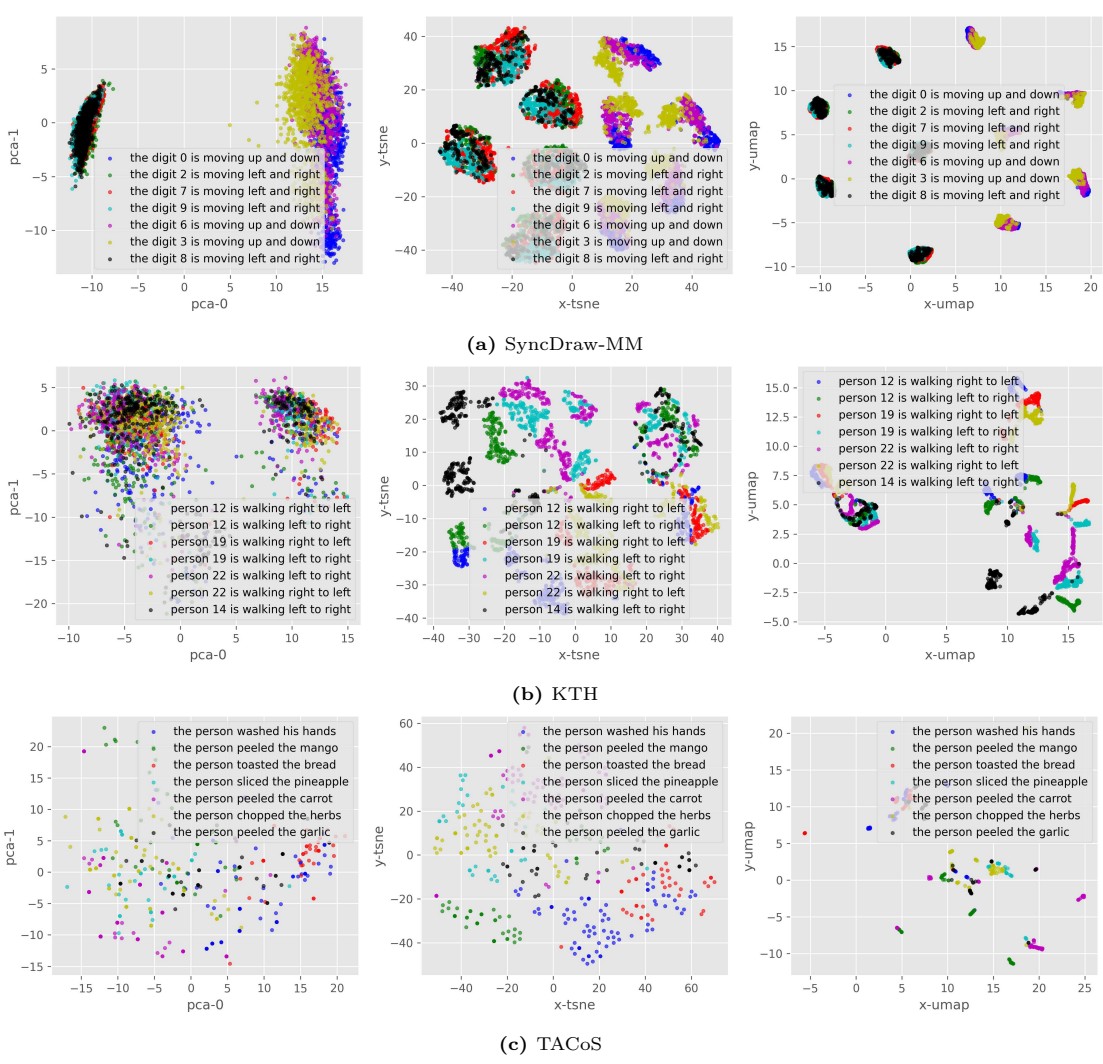

**Figure B.1:** Samples of seven random buckets visualized in the video latent spaces generated using VideoMAE-v2 (Wang et al., 2023) encoder, for each data set. The columns results correspond to: PCA, t-SNE, and UMAP, respectively.

## C.2  Metrics

We used the official implementation of LPIPS (Zhang et al., 2018)[4] and the AlexNet (Krizhevsky et al., 2012) backbone to calculate the metric values. Other parameters were defined with the default values used in the official code. We used the official implementation of DISTS (Ding et al., 2022)[5] and the PyTorch version of the metric. The default backbone used was the one based on VGG16 (Simonyan & Zisserman, 2015) with the default repository parameters.

For distribution-based metrics, we considered the following: FVD is calculated with the I3D video features (Carreira & Zisserman, 2017) extracted from the model (RGB stream) available on Kinects-I3D[6] with an extension of the FID metric from Heusel et al. (2017)[7]; and KVD with the polynomial MMD (Unterthiner et al., 2018).

---

[4] https://github.com/richzhang/PerceptualSimilarity
[5] https://github.com/dingkeyan93/DISTS
[6] https://github.com/google-deepmind/kinetics-i3d
[7] https://github.com/bioinf-jku/TTUR/

### C.3 Architectures

In this section, we present details of each training setup for text, video and the progressive decoupling method. Additionally, Table C.1 shows an overview of the number of parameters of the models used for the video autoencoder, mapping function and video semantic shared autoencoder.

#### C.3.1 Text Models

We evaluated the CLIP (Radford et al., 2021) text encoder, which is used with its pre-trained model from the ViT-B/32 version. The CLIP method used was based on the `transformers` package[8] using the pre-trained model with key `openai/clip-vit-base-patch32` generating a 512-dimensional embedding.

The word dictionary used as the noise set for sampling a noise word for the text in the cross-modal alignment was built based on DBPedia (Lehmann et al., 2015)[9] and is processed similarly to Dai & Le (2015). First, we treat punctuation as separate tokens. Then, we ignore any non-English characters and words. Since the removal of non-English words can affect the semantics of the text, we also remove entries that have too many `UNKNOWN` tokens after this preprocessing. We have defined a maximum value of 45% of unknown tokens to be considered a valid entry for the set. We also remove words that appear only once in the set, and we do not perform any term weighting or stemming in the preprocessing. This word dictionary with the exception of words in each data set is then the final dictionary set used.

#### C.3.2 Video Models

For the video pixel-based discriminator $D_v$, we adapted the Patch discriminator from Pix2PixHD (Wang et al., 2018) that evaluates video quality on multiple scales. For the video representation, we considered the dimensions: $d_z = 64$ for 3DConv-Base and $d_z = 128$ for UnetLDM. Other regularization coefficients were defined as $\lambda_{pixel} = 10$, $\lambda_f = 10$, $\lambda_z = 5$. In particular, we defined $\lambda_p = 0.0025$ since this term dominated other terms in the final loss and this value presented satisfactory results in perceptual quality. In this case, the perceptual weight is defined over the VGG19 layers: `block4_conv3` and `block5_conv4`. The training setup for the video autoencoder considered Adam optimizer with a learning rate of $10^{-4}$ with a global clip norm (maximum gradient norm of 5.0). We trained the video models for about 100 epochs with varying batch size of $32 - 100$, for UnetLDM and 3DConvBase, respectively.

For the video cost in Equation A.2, we found empirically that an L1-based distance converged better for the pixel and feature discriminator terms, while an L2-square distance worked better for perceptual loss.

#### C.3.3 Progressive Decoupling

In the decoupling process, we also consider a second text description input $\hat{t}_i$ from $t_i$, where a noise word is added with probability $p = 0.15$ to include variation in text representation in the same bucket $b_i$, but having the bucket loss considering the original text embeddings $t_i$. The word dictionary from which the noise sample is obtained did not include any words from the corresponding data set corpus. We also evaluated dropout noise (Gao et al., 2021), but empirically found that the addition of random words worked better. Word removal, on the other hand, was not suitable as it directly interferes with the original bucket semantics since removing some words could join samples from originally different buckets.

For the progressive decoupling architecture, we considered a multilayer perceptron (MLP) with four layers. Except for the last layer, each was defined with a hidden layer size of 512 and is followed by a Layer Normalization (Ba et al., 2016) and Swish activation function (Ramachandran et al., 2017). A dropout layer is used after the second and third layers with a rate of 0.1. This was the base network used for the mapping function and video shared AE, changing only the input and output dimensions to match the corresponding representation sizes. The regularization coefficients were defined as $\lambda_{z_s} = 5.0$, $\lambda_s = 100$, $\lambda_{feat} = 10$, $\lambda_{pixel}^s = 30$. In addition, for the bucket loss, we define $\lambda_{neg} = 1.0$, $\lambda_{pos} = 1.0$, and $\alpha = 2.0$ to weight the direct text-video pairs of the bucket. The training setup also considered Adam optimizer with a learning

---

[8] https://huggingface.co/docs/transformers/en/model_doc/clip#transformers.TFCLIPTextModel
[9] Downloaded from https://github.com/srhrshr/torchDatasets/. The data set splits ('train' and 'test') provided were the ones used in our experiments as well.

**Table C.1:** Size of the networks and components used in this work.

| Model | Number of parameters |
| --- | --- |
| 3DConv-Base | 9.9M |
| UnetLDM | 268.8M |
| VDM (Ho et al., 2022) | 35.7M |
| Mapping Function $M$ ($d_{z_t} = 512$ and $d_{z_s} = 64$) | 824k |
| Video semantic shared AE ($E_s$ and $G_s$) ($d_{z_v} = 64$ and $d_{z_s} = 64$) | 1.2M |
| Components | |
| Discriminator $D_s$ or $D_z$ with $d_z = 64$ | 133K |
| Discriminator $D_v$ (PatchHD-Video) | 2.6M |
| VGG19 (Johnson et al., 2016) network | 20M |

rate of $10^{-4}$ with a global clip norm (maximum gradient norm of 4.0). We trained the models for about 70 epochs with varying batch size, which depends on the video autoencoder used and the cross-modality alignment approach, of $32 - 100$.

# D  Additional Results

In this section, we present additional results on the video autoencoder models and the progressive decoupling ablation experiments.

## D.1  Video Representation Learning

In Figure D.1, we present the latent spaces of the 3DConv-Base and UNetLDM video autoencoder models separately, whereas in Section 4.2.1 we presented their joint latent space. Figure D.2 also shows two different runs of PCA, t-SNE and UMAP visualization for the same latent space that exhibits the different overall distribution structures obtained in these different initializations.

In Figure D.3, we present the qualitative results for the video autoencoders: 3DConv-Base, UNetLDM, and VDM (Ho et al., 2022). From the SyncDraw-MM set, we observed better quality with UNetLDM. The 3DConv-Base model generates correct results, but has more misleading cases and lacks sharpness in some cases. In the SyncDraw-MM set, for example, there are cases where digit 5 is misplaced with 3, or 9 with 4, and 1 with 7. This occurs at a lower level in the other models. The VDM model, on the other hand, is not consistent with its results, with its major drawback being the lack of filling in the digit (e.g., holes in some digits) and the thin look in most samples. This model also does not correctly generate the digits in a large part of the samples, generating instead frames with black background and random white points in the border without any digit enclosed.

For the KTH set, UNetLDM also produces sharper videos compared to 3DConv-Base, which in some cases generates an artifact resembling an aura over the person. In this set, UNetLDM appears to produce a brighter background as well. VDM model produces sharper videos and also a large diversity in the samples, but following the results with the previous set, there is a large amount of poor samples generated where there is no movement or person in the video.

Lastly, for the TACoS set, 3DConv-Base generated videos with less fine-grained details. For some cases, this seems to impact the understanding of the movement depicted in the video. The effect of "aura" also occurs in some samples of this set around people. The UNetLDM model generated more blur effects for TACoS and some artifacts resembling "checkboard" artifacts, mostly in brighter parts. The VDM model produced sharper videos for this set, and it was observed that the majority of the samples were generated with people in darker clothes.

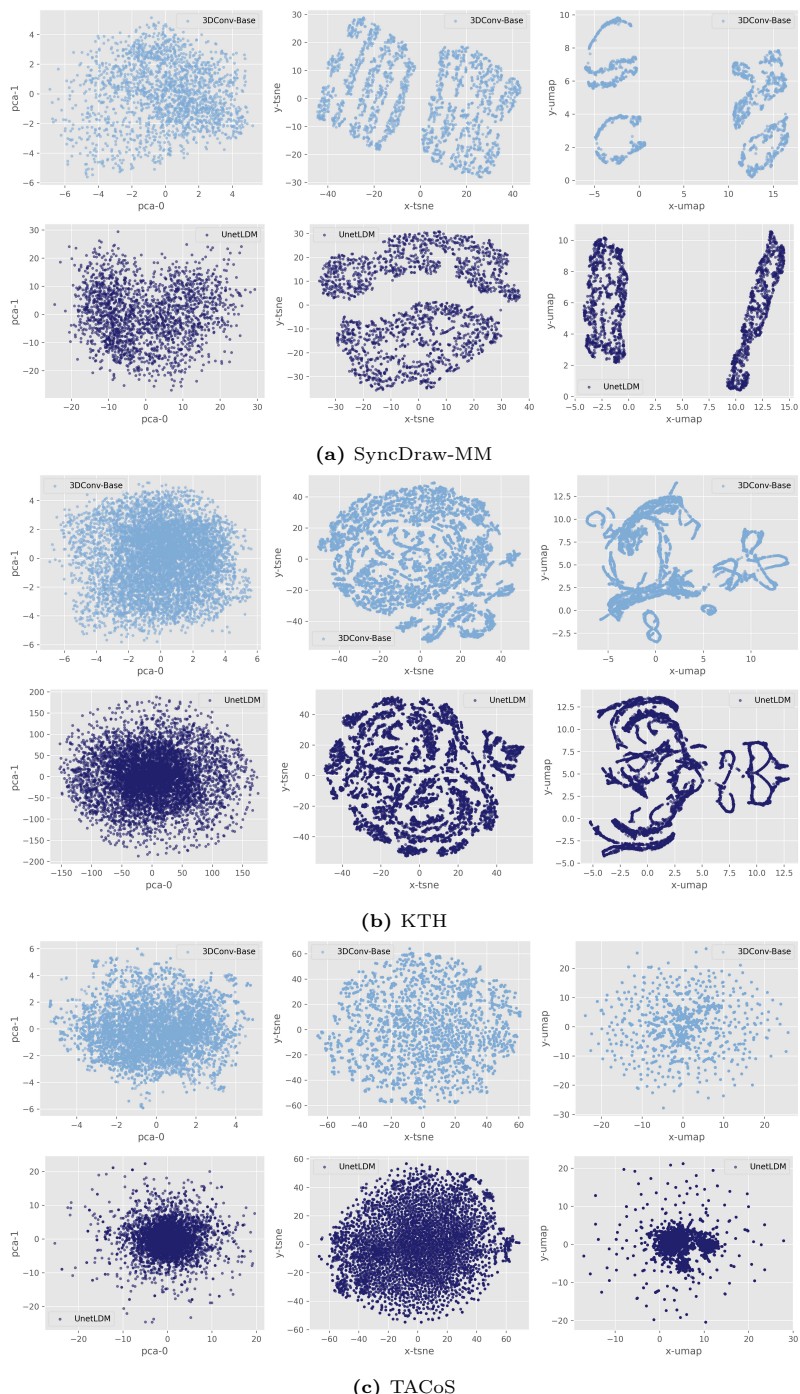

**Figure D.1:** Visualization of the video semantic spaces obtained with the 3DConv-Base (first rows) and UnetLDM (second rows) models for the Syncdraw-MM, KTH, and TACoS data sets.

### D.1.1 Ablation Experiments

Furthermore, we performed ablation experiments on video representation learning, where the quantitative results are presented in Table D.1, qualitative results are shown in Figure D.4, and latent spaces are shown in Figure D.5. We evaluated three main components using the 3DConv-Base architecture: the impact of

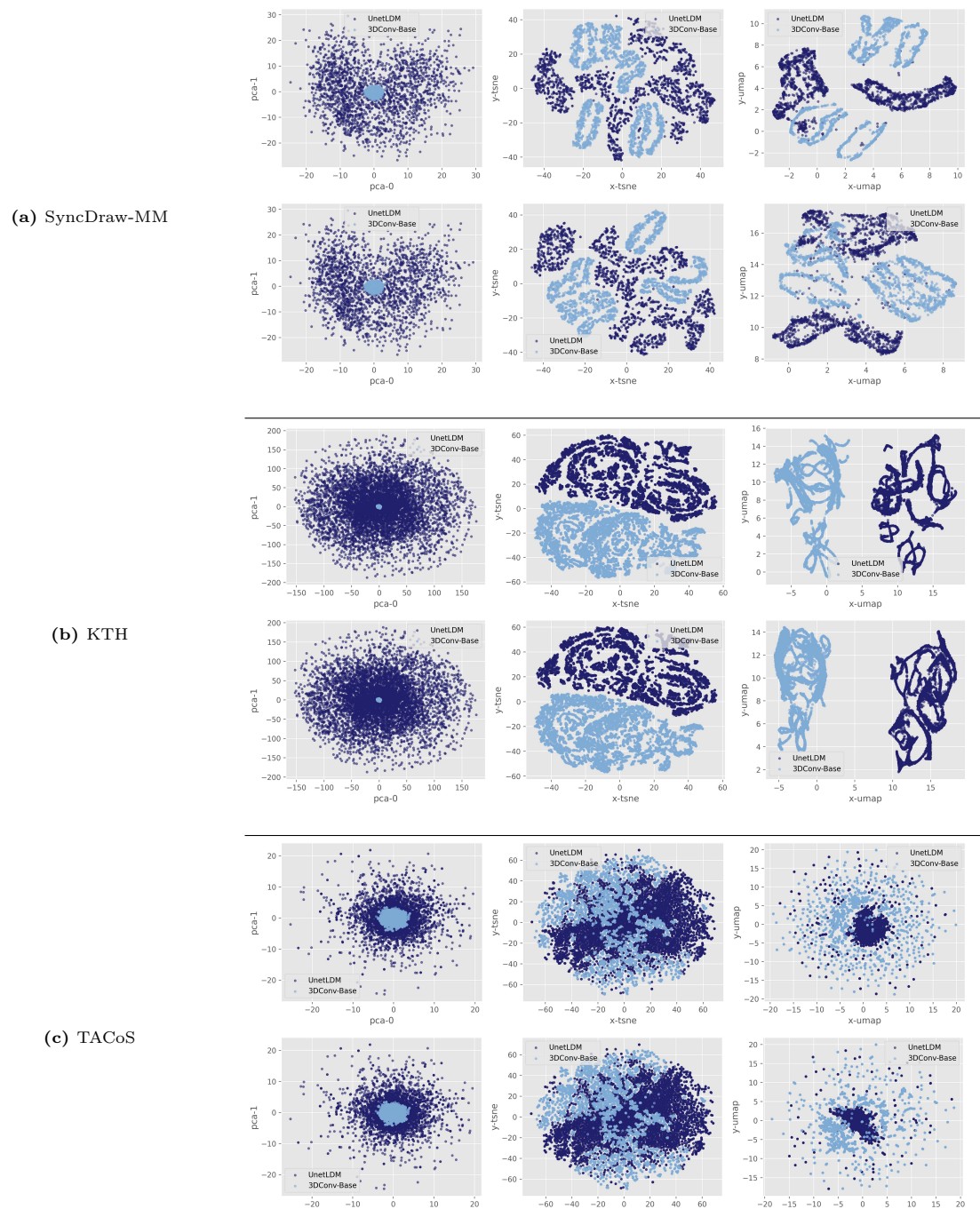

**Figure D.2:** Two runs with different initialization seeds (first and second rows) for the visualization of the video semantic spaces from 3DConv-Base and UnetLDM on the SyncDraw-MM, KTH, and TACoS data sets. For UnetLDM, we first apply PCA to reduce the dimensionality to match the latent dimension of 3DConv-Base. Initialization seed are considered only for t-SNE and UMAP.

the video latent dimension size; the impact of the autoencoder type by comparing with a Variational Auto-Encoder (VAE); and the impact of the distribution discriminator $D_z$ in the WAE-GAN-based approach.

From the quantitative results, we observe that the dimension size variation ($d_z$) does not significantly affect the quantitative results. However, switching from the WAE-GAN-based approach to VAE degrades

**Table D.1:** Quantitative results of the video ablation experiments performed with SyncDraw-MM and the 3DConv-Base video architecture evaluating the impact of: dimension size of latent space $d_z$ and the general autoencoder adopted approach.

| Metrics | PSNR↑ | SSIM↑ | LPIPS↓ | DISTS↓ | FVD↓ | KVD↓ |
|---|---|---|---|---|---|---|
| Dimension | | | | | | |
| $d_z = 48$ | $19.2 \pm 1.9$ | $0.89 \ \pm 0.03$ | $0.083 \pm 0.023$ | $0.09 \pm 0.02$ | 2.81 | 0.0031 |
| $d_z = 128$ | $19.0 \pm 1.8$ | $0.887 \pm 0.030$ | $0.086 \pm 0.024$ | $0.09 \pm 0.02$ | 2.77 | 0.003 |
| $d_z = 256$ | $19.1 \pm 1.9$ | $0.889 \pm 0.030$ | $0.083 \pm 0.023$ | $0.09 \pm 0.02$ | 2.75 | 0.003 |
| General AE approach | | | | | | |
| Base w/o $D_z$ | $18.8 \pm 1.8$ | $0.885 \pm 0.030$ | $0.085 \pm 0.024$ | $0.09 \pm 0.02$ | 2.60 | 0.0028 |
| VAE w/o $D_z$ | $17.3 \pm 1.2$ | $0.860 \pm 0.020$ | $0.112 \pm 0.036$ | $0.11 \pm 0.02$ | 3.34 | 0.0041 |
| VAE | $15.3 \pm 1.2$ | $0.757 \pm 0.040$ | $0.289 \pm 0.066$ | $0.16 \pm 0.02$ | 9.10 | 0.0165 |

reconstruction. Removing the distribution discriminator $D_z$ also degrades performance, except for the VAE approach, which achieves better results without it.

The qualitative results reveal inferior reconstruction with the plain VAE approach where more than one digit appears to be reconstructed, resulting in a mirror effect. Additionally, most videos seem to be concentrated on "up and down" movements rather than "left to right" movements. However, this effect appears partially associated with the distribution discriminator $D_z$, as removing it improves qualitative results on the same instances. When removing $D_z$ from our main approach, we observe a small decrease in some full-reference metrics (e.g., PSNR) but slightly better distribution-based metrics (e.g., FVD). Qualitatively, this difference is minimal, as the videos from both approaches are similar, with both showing some loss of fine-grained details in the digits. Regarding latent dimension size, we observe minor differences in fine-grained details between models on the same instances.

Moreover, examining the video latent spaces in Figure D.5, we observe slight structural differences across models. For dimension variation, latent spaces with $d_z \geq 128$ are sparser than those with $d_z < 128$ (also including $d_z = 64$ in Figure D.1). In contrast, VAE-based approaches produce more concentrated latent spaces when we evaluate their distribution in the PCA visualization. Additionally, removing the distribution discriminator $D_z$ alters the latent space structure compared to the baseline in Figure D.1.

## D.2 Progressive Decoupling Learning

In Figures D.6, D.7, and D.8, we present qualitative results for text-to-video generation produced with the alignment models of Section 4.2.2 for: SyncDraw, KTH, and TACoS data sets.

From the SyncDraw results, we can observe a more difficult task for the alignment. All models seem to present poor video generation with a lack of fine-grained details for the digits. The worst results being the ones with the ImageBind-based and our method with UNetLDM alignments. They present higher indicators of representation collapse, where the former shows vertical movements even when the input text requires horizontal movements, and the latter generates almost the same exact video for different input texts. Overall, better results were observed for vertical movements compared to horizontal ones, although both are equivalently represented in the data set.

For the KTH set, the models present better results, which is possibly related to the decrease of the one-to-many difficult level of this set. We still observe a level of representational collapse for some input texts, but in a lower level than with the SyncDraw set. For CLIP-based and our method with 3DConv-Base, we observe an aura effect in some persons, which was also identified in the video autoencoder results, indicating a propagation of this effect. On the other hand, CoDi and our method with UNetLDM present less of this artifact. For the ImageBind-based alignment, frames with more blur than the other and an artifact resembling the generation of movement shadow in the legs part were noticed.

For the TACoS set, the results improve when compared with the KTH set, strongly indicating a correlation with the difficulty level of the alignment. The only exception being our method with UNetLDM video baseline, as this model, previously found to have the representational collapse problem in latent space,

**Table D.2:** Quantitative results of additional ablation experiments on the feature alignment between text and video modalities on the SyncDraw-MM data set with 3DConv-Base video autoencoder.

| Model/Metrics | B-PSNR↑ | B-SSIM↑ | B-LPIPS↓ | B-DISTS↓ | FVD↓ | KVD↓ |
|---|---|---|---|---|---|---|
| Dimension $d_z = 256$ | $19.7 \pm 1.9$ | $0.864 \pm 0.027$ | $0.27 \pm 0.10$ | $0.14 \pm 0.02$ | 6.98 | 0.006 |

presents here the video generation indicator of this as well. In this set of results, an aura effect is also observed for models based on 3DConv-Base video AE. The best models being CLIP-based alignment and our method with 3DConv-Base. The aura effect seems more prominent in the ImageBind-based alignment, although other methods also use this video autoencoder and this model also present more blur than the others in the overall videos.

### D.2.1 Ablation Experiments

In Table D.2, we present the quantitative results of additional ablation experiment on the cross-modality alignment with the best architecture found in the video representation ablation. Figure D.9 shows the corresponding latent spaces of the video and shared semantic representations. Lastly, in Figure D.10, we present the qualitative results for the ablation experiment.

The additional ablation shows the impact of representation learning in mapping between the modalities. We note that the structures of latent spaces change when we change the way the target modality is represented. Although it is primarily outstanding in shared semantic spaces, the structure is affected by producing sparser spaces (e.g. $d_z = 256$ ).

Regarding the qualitative results, we observed poor generation, indicating a poor alignment level for the SyncDraw data set, which is the most challenging one-to-many scenario. The non-progressive method trained with our adapted loss shows a higher indicator of representational collapse, since their generated videos seem to follow, with small differences, the same outlined video. The non-progressive version with the InfoNCE loss seems to suffer less with the representation collapse, although the video quality still lacks fine-grained details of the digits.

The VicReg seems to work better with particular digits, such as the digit 4. This is similar to BYOL results in this regard. But this solution in some videos seems to be generation details of two digits in the same scene, although in this scenario ground-truth data does not have it. Regarding the experiment with the video AE 3DConv-Base with $d_z = 256$, we observe better fine-grained details for some digits but far away to be considered correctly aligned. In general, one case noticed in the results was that the models seem to correctly follow the target motion, being better at the "moving up and down" category than the "left to right" or vice versa. Considering that the SyncDraw set is balanced in this regard, i.e., the number of videos with the "moving up and down" is close to the number of videos of "moving left to right" (or vice versa), this can show a more difficult alignment in the later large bucket of horizontal movement.

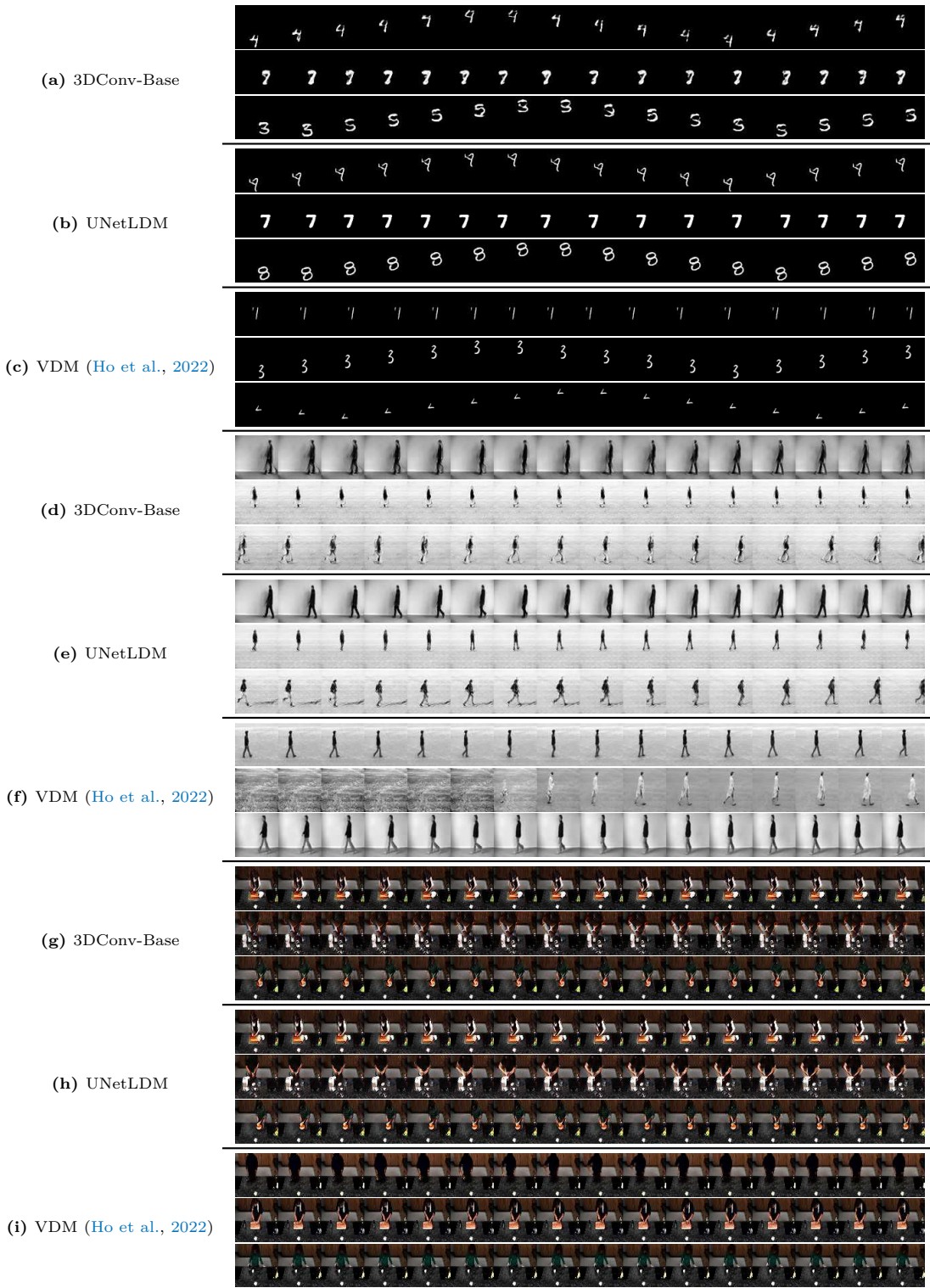

**Figure D.3:** Comparison of the generated videos by the video autoencoder models on the SyncDraw-MM (a-c), KTH (d-f), and TACoS (g-i) data sets with 3DConv-Base, UNetLDM, and VDM (Ho et al., 2022).

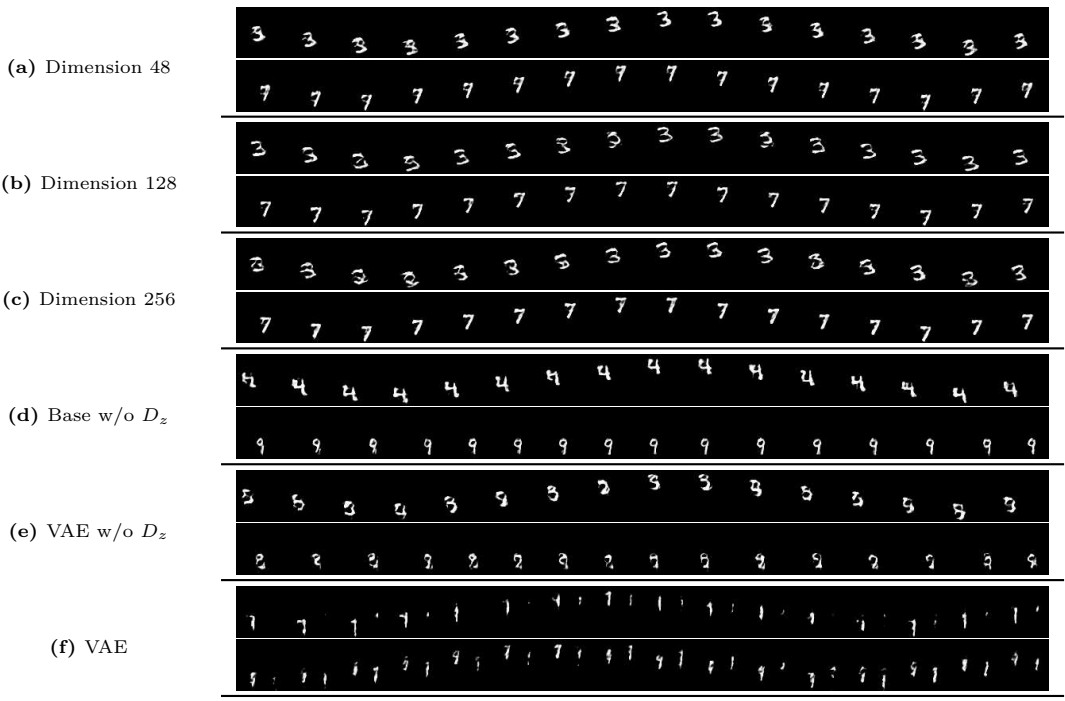

(a) Dimension 48

(b) Dimension 128

(c) Dimension 256

(d) Base w/o $D_z$

(e) VAE w/o $D_z$

(f) VAE

**Figure D.4:** Comparison of the generated videos by the video AE models from the ablation experiments on the SyncDraw-MM data set with 3DConv-Base architecture and variations on latent space dimension and AE approach.

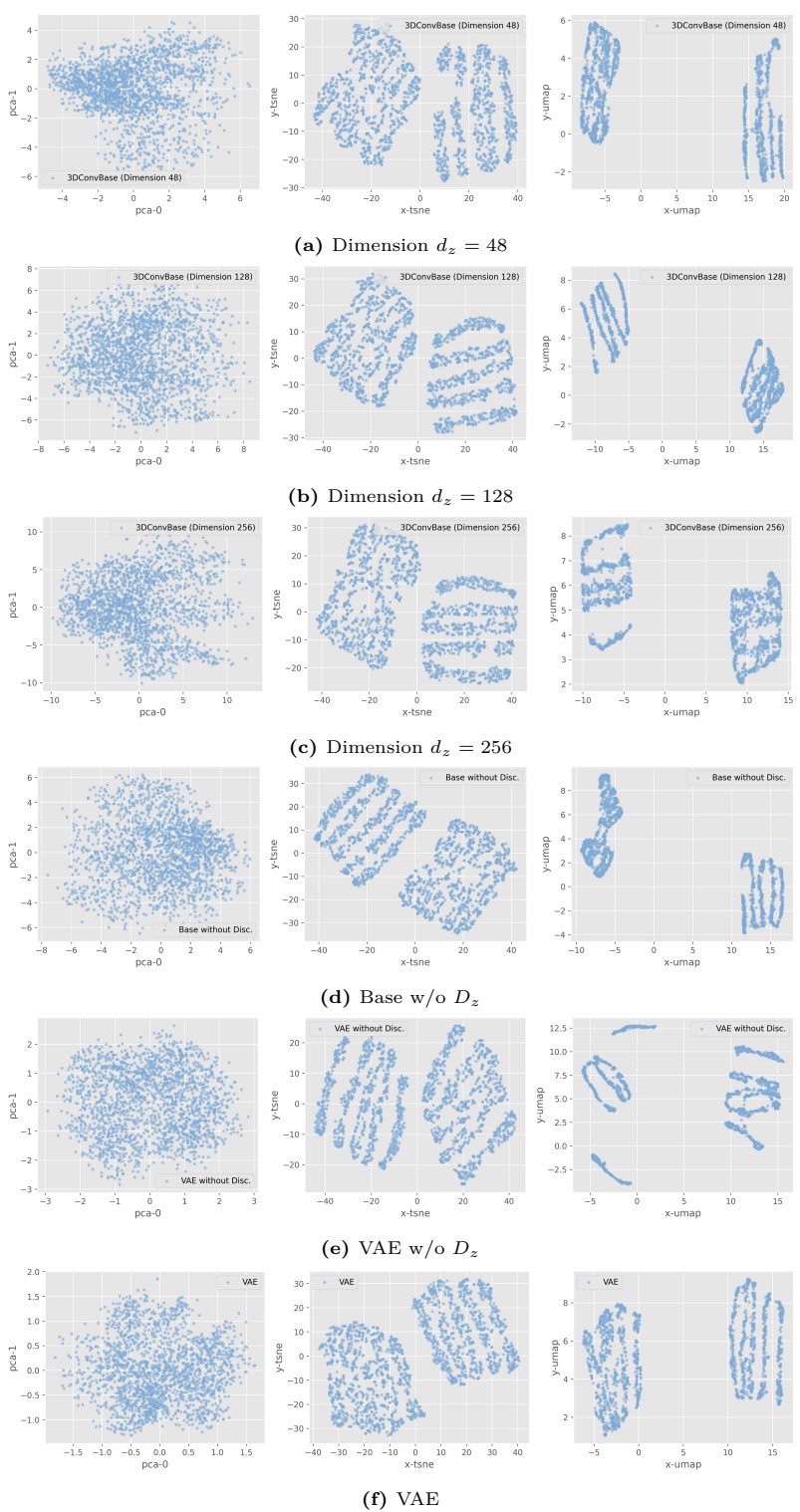

**(a)** Dimension $d_z = 48$

**(b)** Dimension $d_z = 128$

**(c)** Dimension $d_z = 256$

**(d)** Base w/o $D_z$

**(e)** VAE w/o $D_z$

**(f)** VAE

**Figure D.5:** Visualization of the video semantic spaces obtained using 3DConv-Base for video ablation experiments on: (a-c) latent dimension $d_z$, (d) removal of distribution discriminator $D_z$, and (e-f) VAE approach.

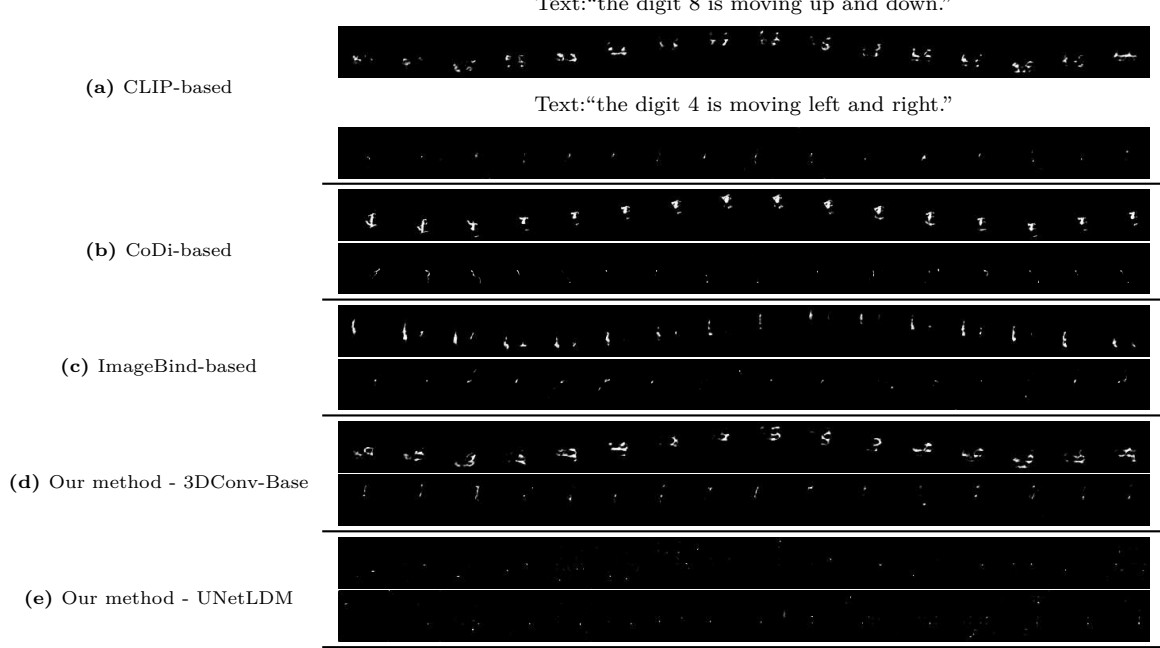

**Figure D.6:** Comparison of the generated videos by the alignment models: CLIP-based, CoDi-based, ImageBind-based, Our method with both 3DConv-Base and UNetLDM video architectures, on the SyncDraw data set.

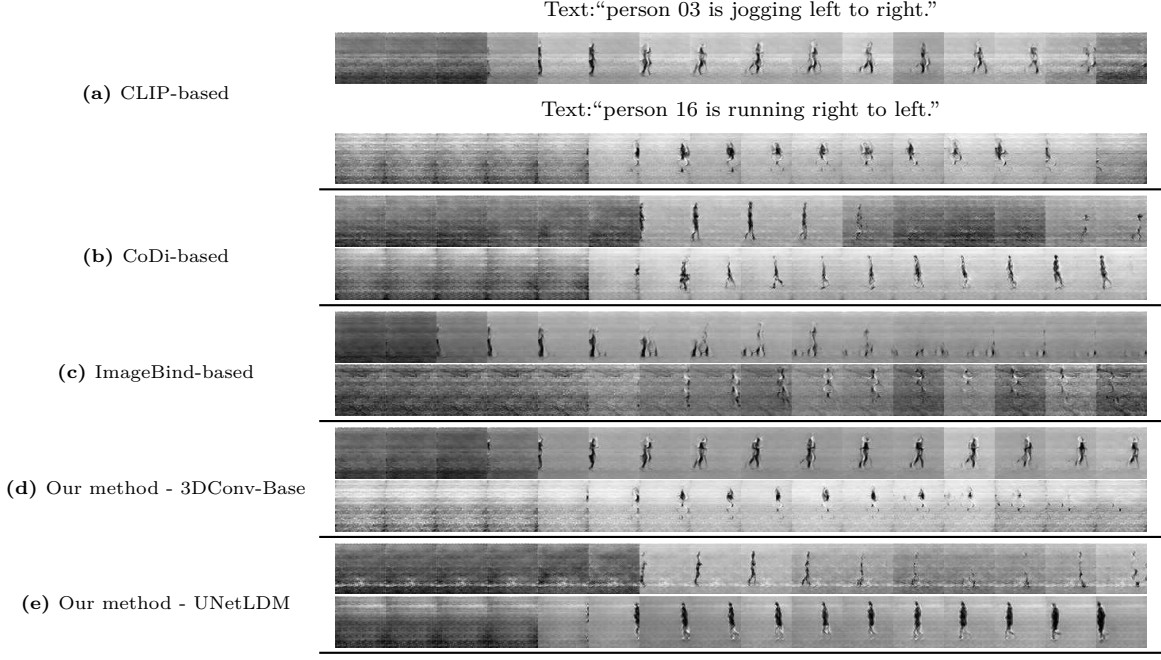

**Figure D.7:** Comparison of the generated videos by the alignment models: CLIP-based, CoDi-based, ImageBind-based, Our method with both 3DConv-Base and UNetLDM video architectures, on the KTH data set.

Text:"the person cut those beans into very small pieces."

Text:"the person cut a plum in half."

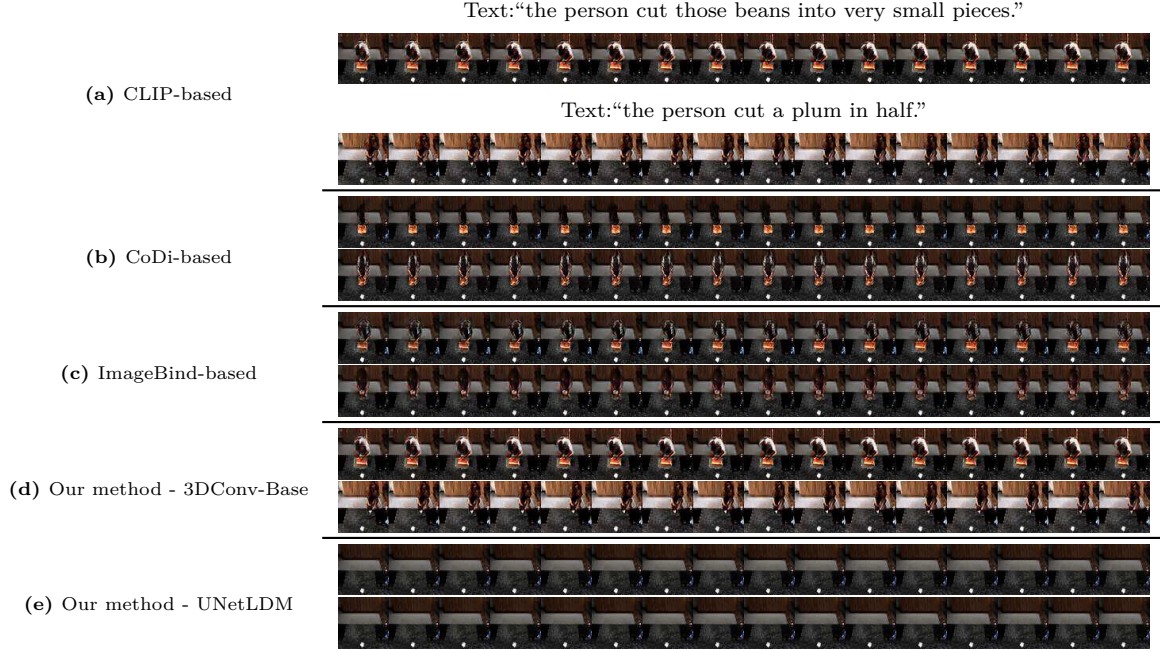

**(a)** CLIP-based

**(b)** CoDi-based

**(c)** ImageBind-based

**(d)** Our method - 3DConv-Base

**(e)** Our method - UNetLDM

**Figure D.8:** Comparison of the generated videos by the alignment models: CLIP-based, CoDi-based, ImageBind-based, Our method with both 3DConv-Base and UNetLDM video architectures, on the TACoS data set.

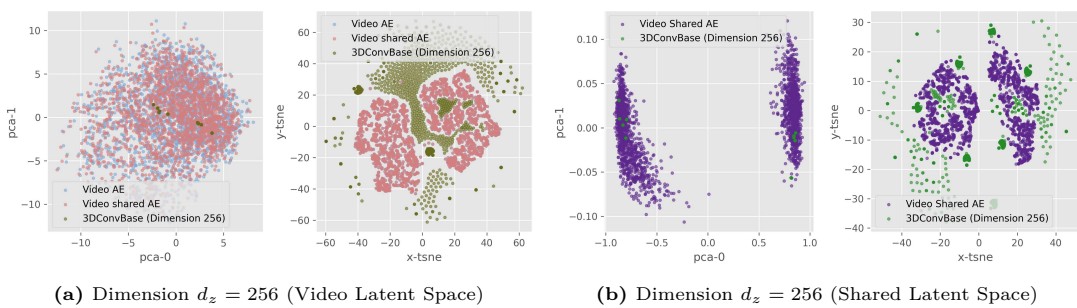

**(a)** Dimension $d_z = 256$ (Video Latent Space)

**(b)** Dimension $d_z = 256$ (Shared Latent Space)

**Figure D.9:** Video latent space (left side) from ablation experiment with video representation learning using $d_z = 256$, showing embeddings from the mapping functions, video shared autoencoders, and video representation learning. Additionally, shared semantic latent space (right side), showing embeddings from the mapping functions and video shared autoencoders, visualized with PCA (odd columns) and t-SNE (even columns).

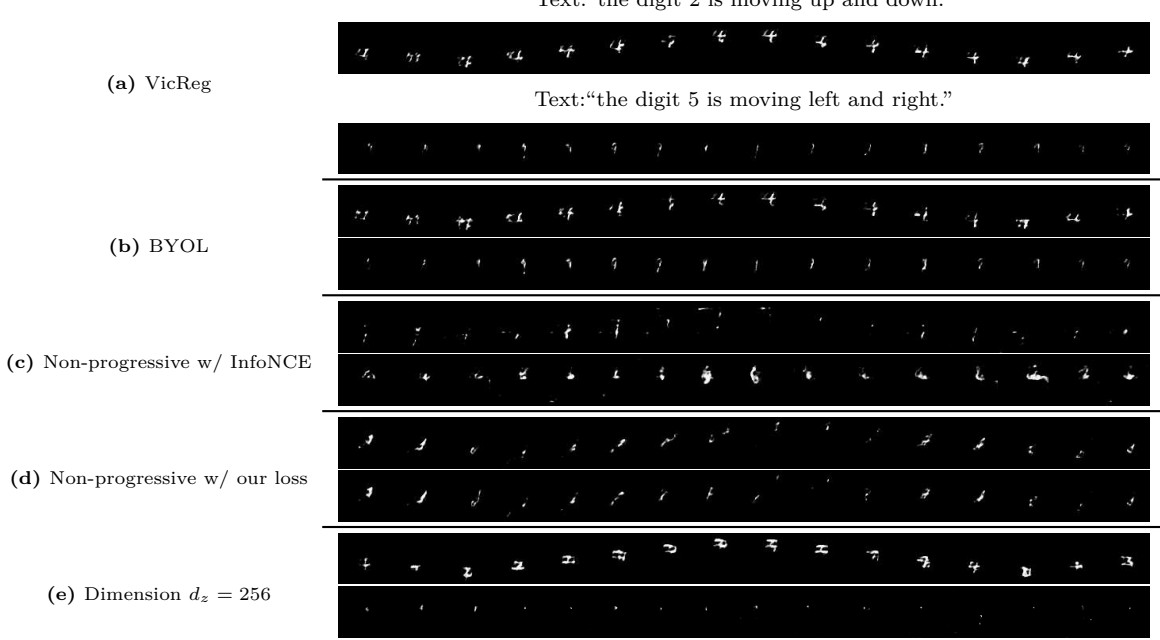

**Figure D.10:** Comparison of the generated videos by the ablation alignment models: VicReg, BYOL, Non-progressive w/ InfoNCE, Non-progressive w/ our loss, and video AE baseline with dimension $d_z = 256$, considering the video 3DConv-Base architecture on the SyncDraw data set.

