# OpenReview forum: "Towards Bridging the Semantic Spaces of the One-to-Many Mapping in Cross-Modality Text-to-Video Generation"
_TMLR — Rejected by TMLR_

### Review · Reviewer_CBUd · 2026-01-03

**Summary Of Contributions:**

The paper presents a pipeline for text-to-video latent space alignment centered around a variation of Wasserstein AEs. The framework is in particular motivated by making use of pre-trained encoders and the authors  study the challenges in this task in the on-to-many mapping scenario, in which one text can be mapped to several valid videos.

The authors compare various backbones (clip Vit-B/ for the text encoder, and for the video encoders  a 3D version of the 2D DCGAN  architecture,a Unet varation adapted from the  2022 latent diffusion model,  and the 2022 VDM), alignment losses (if I read correctly, at least a feature matching divergence and a contrastive  variant?) and datasets (Moving mnist, kth human action,TACoS multi-level corpus), studying both generative modeling fidelities and latent space structure via t-SNE,UMAP and PCA.

Aside from presenting their pipeline and their investigaton of different mappings, variations etc, the authors also claim the identification of the one-to-many mapping scenario as a key challenging case in cross-modality text to video as a key contribution.

**Additional Comments:**

there are also various typos and (to me) weird word choices/phrasings e.g. Wassertein AE on page 4, a stray "nevertheless" conjuction in 2.2. which does not parse well for me, the word "reasoning" used for what to me would be finding or fitting, "more free collapse latent spaces" on page 9, which I assume to mean "more collapse free" or something along the lines? I'd recommend a pass with a lektor (or the poor mans lektor, having the latest llm critique the work and manually deciding what to fix)

**Audience:**

Yes

**Audience Explanation:**

I would be interested, and a high quality study of latent spaces, alignment challenges and variation across them would have value.

**Claims And Evidence:**

No

**Claims Explanation:**

I like investigatory papers like this, but I think as is, it needs a rewrite to streamline the story and organize itself.

- the one to many mapping isn't motivated for being relevant or why the particular measurement (bucketized metrics) are valid operationalisations of it. I know it _is_ relevant for multiple reasons (avoid arbitrarily picking a canonical variant, principled variational modeling _requiring_ being able to map to multiple latent states if the text carries less info than the video, to give two) but neither are explicitly _stated_ in the intro. I think some motivation and visual examples would go a long way in motivating the study
- t-sne is used to illustrate next to PCA and UMAP in what feels like a grab bag, without justification. given that it is stochastic, I fail to see why it is included
 details of ablations, losses, architectures, as well as hyperparameter choices and insights observed in training are strewn across the paper and the appendix (which, to give credit is substantive) with no clear structure. I think _grouping_ things and also defining the losses a to Z in completely typset equations + a glossary table for all the symbols would go a long way in making this more readable (as would organizing variations and their results in tables/pareto figure plots...overall more structured presentation)
-  as noted the bucketized setting  and measurement, as well as the overall framing requires more justification. what is the argument that the one-to-many issue is not simply an issue in the authors particular pipeline? there is extensive liteture discussed (good) but at least I did not find notes of this being a known or well established  (just not yet formally studied) challenge.

if the paper can be organized into clearer research questions and sections ala

1. the problem and why we should care
2. the overall methods we will employ in studying
3. research question 1, the specific variation used to study, the results
4. (repeat for all research questions)
5. conclusions
6. limitations
appendix for details (but not required to be able to read the rest of the text, only completeness)

it would go a long way towards finding value in the paper

**Requested Changes:**

I have laid out my main issues above, the main changes I would make are

1. organize the information into more cohesive units (I should be done learning about the architectures tried after reading one section and at most 1 appendix, then I should not encounter new variations. I should not learn about a new investigation target in the middel of a chapter etc)
2. motivate the framing more, to be able to connect the chosen metrics and ways of studying to clearer statements in relation to that motivation

---

> ### Author Response · Authors · 2026-01-28
> **Response to Reviewer CBUd**
>
> Dear Reviewer CBUd,
>
> First, we want to thank you for your time and work reviewing this manuscript. Please see our responses below and feel free to point out anything that could further improve our work.
>
> > I like investigatory papers like this, but I think as is, it needs a rewrite to streamline the story and organize itself.
> >
> >  - the one to many mapping isn't motivated for being relevant or why the particular measurement (bucketized metrics) are valid operationalisations of it.
> >     I know it is relevant for multiple reasons (avoid arbitrarily picking a canonical variant, principled variational modeling requiring being able to map to multiple latent states if the text carries less info than the video, to give two) but neither are explicitly stated in the intro.
>
> Thanks for pointing this out. We have reviewed the problem motivation throughout the manuscript and clarified our rationale for using bucket-based metrics in text-to-video evaluation. Please refer to the revised manuscript.
>
> > t-sne is used to illustrate next to PCA and UMAP in what feels like a grab bag, without justification. given that it is stochastic, I fail to see why it is included details of ablations, losses, architectures, as well as hyperparameter choices and insights observed in training are strewn across the paper and the appendix (which, to give credit is substantive) with no clear structure.
>
> We have added a discussion of our dimensionality reduction technique choices, as also suggested by Reviewer AkH5. Our main motivation was to use global and local-structure-preserving methods together as complementary approaches. Although t-SNE and UMAP are stochastic, we can set random state variables to reproduce the same visualization. Moreover, we focus not on exact structural replication but rather on distributional relationships. We also include some examples in Appendix D (`Additional Results') to illustrate how the variations are not impacting the interpretation of the relationships presented between them.
>
> Regarding the structure, we have reviewed the manuscript content, primarily in the Experiments and Appendix sections. Please refer to the revised manuscript.
>
> > I think grouping things and also defining the losses a to Z in completely typeset equations + a glossary table for all the symbols would go a long way in making this more readable (as would organizing variations and their results in tables/pareto figure plots...overall more structured presentation)
>
> We didn't understand the losses and equations suggestion. We have checked the losses definitions and we are not sure what could be missing.  Could you clarify your point about the equations? We want to ensure we address any missing components.
>
> > as noted the bucketized setting and measurement, as well as the overall framing requires more justification. what is the argument that the one-to-many issue is not simply an issue in the authors particular pipeline? there is extensive literature discussed (good) but at least I did not find notes of this being a known or well established (just not yet formally studied) challenge.
>
> Thanks for this comment. While cross-modality research is being pursued across several domains, few works focus on generation, particularly text-to-video generation. In the related work section, we present only one prior method that is proposed for text-to-video generation, which is the CoDi model. Most works focus on representation learning, retrieval, or other generation tasks.
>
> Also, note that the data sets used in most text-to-video generation methods are large-scale (millions of pairs). For example, CoDi was trained on WebVid (2M text-video pairs) and HD-Vila-100M (100M text-video pairs). As we mentioned in our response to Reviewer AkH5, such large data sets may simply mask the one-to-many problem rather than address it.
>
> Regarding if the one-to-many problem is an issue with our pipeline, we note that at lower difficulty levels of the problem, we mostly observe better alignment among the methods, suggesting the bottleneck is the one-to-many challenge rather than the method itself. Furthermore, the progressive alignment has similar, though not identical, hierarchical latent space structure to prior work (Xu et al., 2019), which was evaluated in a different alignment context.

---

> > ### Author Response · Authors · 2026-01-28
> > **Following the previous response to Reviewer CBUd**
> >
> > > if the paper can be organized into clearer research questions and sections ala
> > >
> > >    1. the problem and why we should care
> > >    2. the overall methods we will employ in studying
> > >    3. research question 1, the specific variation used to study, the results
> > >    4. (repeat for all research questions)
> > >    5. conclusions
> > >    6. limitations appendix for details (but not required to be able to read the rest of the text, only completeness)
> >
> > Thank you for your constructive suggestion. We have reviewed the manuscript structure along with your other suggestion regarding cohesion. The experiments section was mostly restructured to present its details upfront, the latent space analysis for the ablation experiments of Section 4.2.3 was moved from Appendix to its corresponding section, and finally the Appendix was reorganized. Please refer to the revised manuscript.
> >
> > > I have laid out my main issues above, the main changes I would make are
> > >
> > >   1. organize the information into more cohesive units (I should be done learning about the architectures tried after reading one section and at most 1 appendix, then I should not encounter new variations. I should not learn about a new investigation target in the middle of a chapter etc)
> >
> > Thank you for your constructive suggestion. We have reviewed the manuscript and its overall structure to improve section cohesion. The latent space analysis of the ablation experiments directly related to alignment was moved to the main ablation section of the manuscript. Additionally, the experiments section was restructured to present its details upfront before dividing into four main important sections: (1) overall latent space structure of the one-to-many scenario, (2) impact of different architectures on the video target distribution, (3) alignment strategies to learn a shared space between modalities, and (4) ablation experiments. The appendix was also restructured.
> >
> > > 2. motivate the framing more, to be able to connect the chosen metrics and ways of studying to clearer statements in relation to that motivation
> >
> > Thanks for your constructive suggestion. We have reviewed the motivation, the rationale for our metric choices, and the main objectives of our investigation presented in the manuscript.
> >
> > > there are also various typos and (to me) weird word choices/phrasings e.g. Wassertein AE on page 4, a stray "nevertheless" conjuction in 2.2. which does not parse well for me, the word "reasoning" used for what to me would be finding or fitting, "more free collapse latent spaces" on page 9, which I assume to mean "more collapse free" or something along the lines? I'd recommend a pass with a lektor (or the poor mans lektor, having the latest llm critique the work and manually deciding what to fix)
> >
> > Thank you for pointing out those issues. We have reviewed the manuscript to address such cases and to detect other typos and textual problems. Please see the revised manuscript with the corrections.

---

### Review · Reviewer_J6XA · 2026-01-08

**Summary Of Contributions:**

This paper studies an underexplored but important issue in cross-modality text-to-video generation: the one-to-many mapping problem in feature-level alignment. Instead of focusing on conditioning-based generation, the authors analyze how independently learned text and video latent spaces can be aligned into a shared semantic space without collapse. The work proposes a unidirectional progressive alignment framework built on autoencoder-based representations and introduces a bucket-based loss to explicitly handle one-to-many relationships. Extensive latent space visualizations and controlled experiments on multiple datasets provide insights into how different alignment strategies behave under varying degrees of one-to-many complexity.

**Additional Comments:**

NA

**Audience:**

Yes

**Audience Explanation:**

I think this paper’s target and topic are important topic in the community. The focus on one-to-many mappings addresses a fundamental limitation of contrastive alignment methods that is relevant not only to text-to-video generation but also to broader multimodal learning settings. Some readers would be interested in this paper.

**Claims And Evidence:**

Yes

**Claims Explanation:**

I think the empirical evidence is clear for the one-to-many failure mode. And systematic latent space analysis supports the main diagnostic claims. The experiments are extensive, and anaylsis is proper.

**Requested Changes:**

1. The authors can consider changing some insights: more clearly stating that its main contribution is the analysis and understanding of one-to-many alignment behavior, rather than achieving large performance improvements. In particular, the quantitative results in Table `2` show relatively small differences across methods, suggesting the contribution is primarily diagnostic.
2. While Figures `4` and `5` provide strong evidence of improved latent alignment, these improvements do not always translate into better video generation metrics. A short discussion explaining this discrepancy (*e.g.*, metric limitations or decoder bottlenecks) would help readers better interpret the results.
3. The progressive alignment and bucket-based loss are central to the method, but their motivation is currently presented in a fairly technical manner. Adding a brief, more intuitive explanation (supported by the ablation results in Table `3`) would make the design choices easier to understand.

---

> ### Author Response · Authors · 2026-01-28
> **Response to Reviewer J6XA**
>
> Dear Reviewer J6XA,
>
> First, we want to thank you for your time and work reviewing this manuscript. Please see our responses below and feel free to point out anything that could further improve our work.
>
> > 1. The authors can consider changing some insights: more clearly stating that its main contribution is the analysis and understanding of one-to-many alignment behavior, rather than achieving large performance improvements. In particular, the quantitative results in Table 2 show relatively small differences across methods, suggesting the contribution is primarily diagnostic.
>
> Thank you for your constructive suggestion. We have reviewed our main contributions statements in the manuscript along with the ones related to Table 2.
>
> As also pointed in an response for Reviewer AkH5, we agree that the quantitative metrics between our proposed framework and some other methods are tight. We propose that additional assessment, such as the latent space perspective serves as complementary information in this case. When evaluating the alignment and qualitative part, we found conflicting results for some methods, which were able to be identified in the latent space analysis.
> We show that an assessment with these two perspectives allows a better evaluation of current alignment methods. Although not reaching the best performance in all cases, our proposed framework analysis is directed to the one-to-many case, without imposing restrictions for a one-to-one case as well.
>
> > While Figures 4 and 5 provide strong evidence of improved latent alignment, these improvements do not always translate into better video generation metrics. A short discussion explaining this discrepancy (e.g., metric limitations or decoder bottlenecks) would help readers better interpret the results.
>
> Thank you for your constructive suggestion. We added a brief discussion explaining this discrepancy in last paragraph of Section 4.2.2. One of the main bottlenecks found was the capacity of the video decoder to decode from regions near but not within the true distributions. Although the alignment could generate close approximations between generated and true distributions, the generated distribution could be in regions unknown to the decoder. Thus, making similar incorrect results achieve similar metrics, which is better evaluated with the latent space perspective in our investigation. However, as we mention in the manuscript this also has limitations, as it is limited to either a global distribution analysis or manual check of small regions of the distribution.
>
> > The progressive alignment and bucket-based loss are central to the method, but their motivation is currently presented in a fairly technical manner. Adding a brief, more intuitive explanation (supported by the ablation results in Table 3) would make the design choices easier to understand.
>
> Thank you for your constructive suggestion. We have added additional discussion of the progressive alignment and the bucket loss in Section 4.2.3 (Ablation). Initial explanations of these design choices are also provided in Section 3 ("Bridging the Semantic Spaces by Progressive Decoupling"). Our investigation revealed that progressive alignment achieves better alignment than non-progressive approaches, although some challenges remain. In contrast, one-phase alignment generally collapsed in early training stages across various hyperparameter settings in our experiments. We found greater stability with the progressive approach.
>
> Moreover, since most methods do not assume the one-to-many case, the text-video pairs are treated as one-to-one references. To encourage alignment within the bucket and maintain intra-bucket samples further apart, we consider a bucket-based loss. Additionally, with lowest difficulty cases of the one-to-many scenario, this loss resembles standard contrastive losses that encourage only the direct match in the alignment.

---

### Review · Reviewer_AkH5 · 2026-01-13

**Summary Of Contributions:**

The paper investigates how the latent embeddings of text and video relates. The authors have observed that simple joint embeddings, using CLIP and the like, do address the fact that a single text may be semantically aligned with mutiple videos. They propose a hierarchical model that takes one VAE for text, and one for video, and aligns then in a shared, semantic, representation space. They propose a loss function to train this model. They analyze how different the embeddings are affacted by model coices, training methods, and the datasets themselves.

**Additional Comments:**

I don't think PCA was super informative. And while you state on page 5 that you use UMAP, but you barely use it. I think you should motivate the choice of dimension reduction technique and just use one to simplify the presentation.

figure 4 is quite complicated. can you simplify somehow?

**Audience:**

No

**Audience Explanation:**

As inficated in the answer to the previous question, I did not find the results compelling enough. The results are a little too vague, and the benefit over alternatives are not clear.

**Claims And Evidence:**

Yes

**Claims Explanation:**

yes. their claims are
- We identify…
- We propose…
- We investigate…

and indeed, they have identified, proposed and investigated. they have identified that the one-to-many mapping problem could be an issue (I don't think they have shown it is a problem in practise). they have proposed a method to train aligned models (but it performs on par with more naive methods). and they investigate the embedding method (but it is not clear what actions should be taken based on the insights herein)

**Requested Changes:**

I think there is a good paper in here, but it is a little too unfocussed. I recommend you to focus on the one-to-many problem, and show the superiority of the bucket based training method on some dataset. Experiments leading up to table 2 and table 3 are quite nice. maybe expand them? in table 2, you highlight rows based on the point estimate, but overlap is significant. I think you need a case where 'our method' clearly provides a benefit.

When you have shown there are *some* cases where the one-to-many mapping is a problem, I think you comprehensive review is interesting for showing the strengths and weaknesses of the method,

---

> ### Author Response · Authors · 2026-01-28
> **Response to Reviewer AkH5**
>
> Dear Reviewer AkH5,
>
> First, we want to thank you for your time and work reviewing this manuscript. Please see our responses below and feel free to point out anything that could further improve our work.
>
>
> > yes. their claims are
> >
> >    - We identify…
> >    - We propose…
> >    - We investigate…
> >
> > and indeed, they have identified, proposed and investigated. they have identified that the one-to-many mapping problem could be an > issue (I don't think they have shown it is a problem in practise). they have proposed a method to train aligned models (but it performs > on par with more naive methods). and they investigate the embedding method (but it is not clear what actions should be taken based > on the insights herein)
>
> Thanks for your constructive comment. As an investigative approach, the main action would be to treat this case in alignment methods for text-to-video generation. The alignment itself could not be the key bottleneck in this scenario, that is why it is important to understand how this scenario could impact such approaches.
> Moreover, some methods do not rely purely on modality representation alignment -- for example, the original work of CoDi bypasses direct alignment by injecting each modality into another representation learning phase. By adopting a purely alignment-focused perspective, we can identify implicit issues that were previously unaddressed. Additionally, a latent space perspective aids in the analysis, since current evaluation protocols for alignment rely either on video quality metrics and/or alignment of input text semantics with the video generated.
>
> As data sets become increasingly large, reaching millions of text-video pairs, we may simply be masking this problem rather than addressing it. Smaller data sets are valuable for identifying such cases and are underexplored in alignment for text-to-video generation.
>
>
> > I think there is a good paper in here, but it is a little too unfocused. I recommend you to focus on the one-to-many problem, and show the superiority of the bucket based training method on some dataset.
>
> Thanks for your comment and suggestion. We have reviewed the manuscript to focus on the one-to-many analysis, also following Reviewer J6XA's suggestion to clearly state the main contribution as the analysis and understanding of the one-to-many scenario rather than achieving significant performance improvements. We have also restructured the manuscript to create more cohesive sections, as Reviewer CBUd suggested. Please refer to the revised manuscript.
>
> > Experiments leading up to table 2 and table 3 are quite nice. maybe expand them?
>
> Thanks for your constructive suggestion. We did perform additional experiments on alignment methods, but given feedback that the manuscript was somewhat unfocused and suggestions to improve its structure and section cohesion, we decided to not add them in this manuscript. We were concerned that including them might reinforce this perception and introduce confusion.
>
> > in table 2, you highlight rows based on the point estimate, but overlap is significant. I think you need a case where 'our method' clearly provides a benefit.
> >
> > When you have shown there are some cases where the one-to-many mapping is a problem, I think you comprehensive review is interesting for showing the strengths and weaknesses of the method.
>
> We agree that the quantitative metrics between our proposed framework and some other methods are tight. We propose that additional assessment, such as the latent space perspective serves as complementary information in this case. When evaluating the alignment and qualitative part, we found conflicting results for some methods, which were able to be identified in the latent space analysis.
>
> We show that an assessment with these two perspectives allows a better evaluation of current alignment methods. Although not reaching the best performance in all cases, our proposed framework analysis is directed to the one-to-many case, without imposing restrictions for a one-to-one case as well.
>
> As we noted in our response to Reviewer CBUd, we observe that alignment methods perform better at lower difficulty levels of the one-to-many case, suggesting that the fundamental bottleneck is the one-to-many challenge rather than any particular method.
> Overall, this challenge is more prominent for data sets where the input text lacks detailed descriptions - for instance, extensive scene-specific information. Generic descriptions are more likely to be semantically similar to other descriptions.
> Since the current literature uses mostly millions of text-video pairs, this scale may help mask this problem rather than shed light on it.

---

> > ### Author Response · Authors · 2026-01-28
> > **Following the previous response to Reviewer AkH5**
> >
> > > I don't think PCA was super informative. And while you state on page 5 that you use UMAP, but you barely use it. I think you should motivate the choice of dimension reduction technique and just use one to simplify the presentation.
> >
> > Thanks for your comment. We use PCA as a global-structure-preserving method, since t-SNE and UMAP are locally oriented. We have also added a discussion of these choices in Section 4.1. Finally, we moved UMAP results to the Appendix, as we consider it an alternative of a local-structure-preserving technique.
> >
> > > figure 4 is quite complicated. can you simplify somehow?
> >
> > Figure 4 analyzes the alignment of three distributions: (1) the target distribution from video autoencoder (video representation learning phase), (2) the distribution from the video shared autoencoder (video embeddings mapped to shared space and back to video embeddings space), and (3) the generated distribution from the alignment method. We present them together to better visualize their alignment.
> > Do you think splitting this into two separated figures, each showing only two distributions at a time, would improve clarity?

---

> > ### Comment · Reviewer_AkH5 · 2026-02-03
> >
> > Dear Authors,
> >
> > Thank you for listening to feedback, and further developing your work. I have observed updates to the structure, e.g. in section 4.1. These improvements helped me following the structure better. Thank you.
> >
> > I still do not feel that you have demonstrated that the one-to-many mapping is a "key challenge in cross-modality text-to-video generation" which is a claim of you manuscript. I agree it is a problem in principle, but I am not convinced it is a problem in practice. Your bucket loss should improve this issue, but you note in ablations that "This more distant alignment in non-progressive methods is not improved by the bucket loss from Section 3.". So do the bucket loss matter?
> >
> > You comment on "this scale may help mask this problem rather than shed light on it" is insightful and present another hypthesis to pursue in research: is the one-to-many problem data dependent, and and goes away as data set size increase? In what applications do we have to care about the problem at all? I feel this would lie outside the scope of the manuscript currently under review, however.

---

### Decision · Action_Editor_p5NR · 2026-02-22

**Recommendation:** Reject

**Audience:**

No

**Audience Explanation:**

While the one-to-many mapping is a theoretical curiosity, the reviewers shared a concern that the paper lacks its relevance to a practical challenge in modern text-to-video generation. Given the issues identified in the paper appear mainly in small-scale dataset, the issues may likely be mitigated by the scale of larger scale datasets. Thus the findings lack the broader interest required for the TMLR audience.

**Claims And Evidence:**

Yes

**Claims Explanation:**

The paper investigates the one-to-many mapping problem through diagnostic analysis and latent space visualization. The evidence regarding the proposed bucket-based loss is mixed. Reviewers noted that quantitative benefits over standard methods are marginal.